# A blood- and brain-based EWAS of smoking

Aleksandra D. Chybowska [1], Elena Bernabeu [1], Paul Yousefi [2,3,4], Matthew Suderman [2,3,4], Robert F. Hillary [1], Richard Clark [5], Louise MacGillivray[5], Lee Murphy [5], Sarah E. Harris [6], Janie Corley [6], Archie Campbell [1,7], Tara L. Spires-Jones [8,9], Daniel L. McCartney [1], Simon R. Cox [6,10], Jackie F. Price[7], Kathryn L. Evans[1] & Riccardo E. Marioni [1] ✉

DNA methylation offers an objective method to assess the impact of smoking. In this work, we conduct a Bayesian EWAS of smoking pack years ($n = 17,865$, ~850k sites, Illumina EPIC array) and extend it by analysing whole genome data of smokers and non-smokers from Generation Scotland ($n = 46$, ~4–21 million sites via TWIST and Oxford Nanopore sequencing). We develop mCigarette, an epigenetic biomarker of smoking, and test it in two British cohorts. Results of brain- and blood-based EWAS ($n_{brain}$=14, $n_{blood} = 882$, >450k sites, Illumina arrays) reveal several loci with near-perfect discrimination of smoking status, but which do not overlap across tissues. Furthermore, we perform a GWAS of epigenetic smoking, identifying several smoking-related loci. Overall, we improve smoking-related biomarker accuracy and enhance the understanding of the effects of smoking by integrating DNA methylation data from multiple tissues and cohorts.

Cigarette smoking remains a leading cause of preventable death and disease, accounting for approximately 8 million global deaths annually[1]. It is a major risk factor of more than 50 diseases including cardiovascular disease, lung cancer and dementia[2]. As smoking history is often used in clinical risk stratification assessments, enhancing the accuracy of cumulative tobacco consumption measurements has the potential to improve prevention and treatment of smoking-related diseases.

Traditionally, tobacco use has often been quantified using self-report questionnaire-response data, such as pack years or indication of current smoking status (i.e. current, former, never smoker), which are prone to recall bias and do not account for passive smoke exposure[3]. A more objective approach to assess smoking is by measuring the concentration of tobacco-related chemicals. However, the commonly used nicotine biomarker, cotinine, has an average half-life of 15–20 hours[4]. Consequently, the concentration of serum cotinine does not inform about time since cessation in recent quitters and it cannot help to distinguish former smokers from never smokers. This limitation is especially pertinent when estimating the risk of diseases that take many years to develop, such as cardiovascular disease.

Blood-based DNA methylation patterns show great promise as a long-term biomarker of smoking[5]. DNA methylation (DNAm) is a cell- and tissue-specific epigenetic modification of DNA molecules that does not change the DNA sequence itself. It involves the addition of a methyl group to cytosine residues and occurs predominantly at cytosine-phosphate-guanine dinucleotides, also known as CpG sites. CpG methylation levels reflect not only smoking status but also cumulative tobacco exposure and can be informative of time since quitting in former smokers[6–8]. In the majority of CpG sites, smoking-related DNAm changes are dose-dependent and reversible after cessation[9].

[1]Centre for Genomic and Experimental Medicine, Institute of Genetics and Cancer, University of Edinburgh, Edinburgh EH4 2XU, UK. [2]Medical Research Council Integrative Epidemiology Unit at the University of Bristol, University of Bristol, Bristol, UK. [3]NIHR Bristol Biomedical Research Centre, University Hospitals Bristol and Weston NHS Foundation Trust and University of Bristol, Bristol BS8 2BN, UK. [4]Population Health Science, Bristol Medical School, University of Bristol, Bristol, UK. [5]Edinburgh Clinical Research Facility, University of Edinburgh, Western General Hospital, Edinburgh EH4 2XU, UK. [6]Lothian Birth Cohorts, Department of Psychology, The University of Edinburgh, Edinburgh, UK. [7]Usher Institute, University of Edinburgh, 5-7 Little France Road, Edinburgh EH16 4UX, UK. [8]Centre for Discovery Brain Sciences, University of Edinburgh, Edinburgh, UK. [9]UK Dementia Research Institute, University of Edinburgh, Edinburgh, UK. [10]Scottish Imaging Network, A Platform for Scientific Excellence (SINAPSE) Collaboration, Edinburgh, UK. ✉e-mail: riccardo.marioni@ed.ac.uk

In previous studies, blood-based DNAm biomarkers of smoking almost perfectly discriminated smokers from never smokers[7,10]. However, their ability to differentiate former smokers from never smokers was relatively modest. Additionally, most of these studies relied on whole-blood DNAm assessments using arrays, which only measure a pre-selected subset of CpG sites present in the epigenome. For example, the current largest epigenome wide association study (EWAS) of smoking in adults ($n$ ⁓ 15,907) was a meta-analysis conducted using the Illumina 450k BeadChip array ($n$ ⁓ 450,000 CpG sites)[6]. The largest EWAS of smoking, in which DNAm was measured with Illumina EPIC array ($n$ ⁓ 850,000 CpG sites, approximately 5% of all sites on the epigenome) considered blood samples of 15,014 individuals[11].

In this work, we update the existing biomarkers of smoking by analysing whole-blood DNAm levels measured with Illumina EPIC array in 17,865 individuals. For a subset of 46 individuals, we implement a high-resolution approach (⁓ 4 million CpG sites, TWIST human methylome panel and ⁓21 million sites, Oxford Nanopore Sequencing) aimed at measuring methylation levels at CpG sites which are currently absent from arrays. Furthermore, we develop a smoking biomarker (mCigarette) using the EPIC dataset and investigate its associations with self-reported smoking in two external cohorts (Lothian Birth Cohort – LBC1936 and The Avon Longitudinal Study of Parents and Children - ALSPAC). We also investigate variations in methylation patterns in both blood and brain, with DNA methylation measured across five *post-mortem* brain regions in 14 individuals. Finally, we compare the epigenetic proxy of smoking to self-reported smoking by considering the genetic loci associated with these phenotypes in genome-wide association studies (GWASs).

## Results

### EWAS of smoking

First, we ran a Bayesian EWAS of smoking (Fig. 1). There were 17,865 Generation Scotland (GS) volunteers with a measure of pack years of smoking and Illumina EPIC DNAm data. The average age of the sample was 47.6 years (standard deviation [SD] = 14.9), and 59.1% of the participants were female (Supplementary Data 1). In the BayesR EWAS, DNAm explained 50.0% (95% Credible Interval 46.0 – 53.9) of the variance in the pack years phenotype.

Forty-two independent CpGs were associated with smoking at posterior inclusion probability (PIP) > 80%, with 26 of these associations reaching a PIP > 95% (Supplementary Data 2). Among the associations with PIP > 80%, 33 had previously been reported in the EWAS catalogue[12] at $P < 1×10^{-4}$, with 30 of these reaching $P < 1×10^{-7}$. This catalogue is a resource that curates findings from published EWAS studies.

Associations unreported in the EWAS catalogue as being associated with smoking included nine sites at PIP > 80% and two CpGs with PIP > 95%. The former group included intergenic CpGs linked to neurodevelopment and addiction, such as cg22454588 (annotated to *SCAMP5*), cg27110277 (*FGF20*), and cg19404444 (*SKI*). The latter, high confidence associations, included cg02517189 (*GRIK5*) and cg00562553 (*HOXA4*).

### High resolution EWASs of smoking

Next, we extended this analysis by running high resolution EWASs of smoking on a subset of 23 pairs ($n = 46$) of current vs never smokers with Illumina EPIC array (⁓ 850k CpG sites), TWIST human methylation panel (⁓ 4 million CpG sites), and Oxford Nanopore Technologies (ONT) sequencing data (⁓ 21 million CpG sites). At $P < 3.6×10^{-8}$ (significance threshold set as per Saffari et al.[13]), the EPIC-based analysis revealed 15 CpG sites associated with smoking status (EWAS inflation factor, $λ = 0.94$), while the TWIST-based and ONT-based analyses identified 33 ($λ = 1.60$) and 9 ($λ = 1.11$) associations, respectively. At a less stringent threshold ($P < 1×10^{-5}$), these counts increased to 42, 102, and 63 for the EPIC-, TWIST-, and ONT-based analyses, respectively. The overlap between the sites identified by these technologies is

detailed in Supplementary Data 3. Figure 2 and Supplementary Fig. 1 (comparison of beta estimates) display the results obtained from the TWIST, ONT and EPIC EWAS.

Among the 33 associations identified as significant in the TWIST EWAS at $P < 3.6×10^{-8}$, two had been previously reported in the EWAS catalog (based on DNAm profiled with array technologies). These included *AHRR* (chr5-373263-373264, beta = −0.35, $P = 1.2×10^{-10}$) and an intergenic locus found on chromosome 2 (chr2-232419951-232419952, beta = −0.24, $P = 3.5×10^{-8}$). The remaining 31 loci significant at this threshold included sites annotated to *F2RL3* (chr19-16889741-16889742, beta = -0.31, $P = 1.3×10^{-11}$) and *USP42* (chr7-6126706-6126707, beta=0.05, $P = 1.8×10^{-8}$). At a less stringent threshold of $P < 1×10^{-5}$, 98 uncatalogued sites were identified. They included *SST* (chr3-187670342-187670343, beta = −0.09, $P = 2.2×10^{-7}$) and *TSPAN5* (chr4-98472405-98472406, beta = -0.06, $P = 6.8×10^{-6}$). Further details are provided in Supplementary Data 4.

In the ONT EWAS, 9 sites were significant at $P < 3.6×10^{-8}$, of which only one had been previously listed in the EWAS catalog: a site mapping to *AHRR* (chr5-373263-373264, beta = -0.47, $P = 2.4×10^{-12}$). The remaining eight included additional loci within the *AHRR* region, loci from an intergenic region on chromosome 2 (e.g., chr2-232420079-232420080, beta = -0.37, $P = 3.4×10^{-10}$) and a site annotated to *CNTNAP2* (chr7-147245588-147245589, beta=0.37, $P = 1.1×10^{-8}$). At $P < 1×10^{-5}$, the ONT analysis revealed 62 uncatalogued loci, such as *SEPTIN9* (chr17-77351321-77351322, beta = -0.27, $P = 8.4×10^{-6}$) and *TERF2* (chr16-69398923-69398924, beta=0.12, $P = 8.7×10^{-6}$). Complete results are available in Supplementary Data 5.

A gene set enrichment analysis of genes mapped to the 102 CpGs with $P < 1×10^{-5}$ identified in the TWIST EWAS revealed 13 enriched gene sets (FDR $p < 0.05$; see Supplementary Data 6). These included tissue degradation through altered extracellular matrix dynamics and chronic inflammation stemming from dysregulated ATP release and immune cell recruitment. Many of the enriched pathways were driven by the presence of collagen genes, such as *COL4A4* and *COL4A3*. In contrast, enrichment analysis based on the significant CpGs from the ONT EWAS did not identify any significantly enriched gene sets.

### DNAm biomarker of cigarette consumption - mCigarette

A DNAm biomarker of cigarette consumption, mCigarette, was then developed. The biomarker was trained in GS ($n = 17,865$) using elastic net regression with 10-fold cross-validation. Prior to training, CpG sites were prefiltered to those associated with tobacco use at FDR < 0.05 ($n_{CpG}$=18,760) in the previous largest EWAS of smoking[6]. This meta-analysis did not include GS and was conducted using Illumina 450k BeadChip array. Following 10-fold cross validation, an optimal lambda value that minimised the mean prediction error was selected (lambda=0.012577) and fed into an elastic net model of smoking pack years. As a result, non-zero coefficients were assigned to 1,255 CpG sites (Supplementary Data 7).

The biomarker was tested in the external LBC1936 study ($n = 882$, mean age 69.6 years, SD = 0.8). Assessing the predictive performance using Area Under the Curve (AUC) (Fig. 3) revealed very good (AUC = 0.85) to near perfect (AUC = 0.98) ability to distinguish between current ($n = 101$), former ($n = 368$) and never smokers ($n = 413$). Similar results were obtained when Area Under the Precision-Recall Curve was used as a performance metric (PRAUC $_{current\ vs\ never}$ = 0.96, PRAUC $_{former\ vs\ never}$ = 0.85, PRAUC current vs former = 0.71).

The predictive performance of mCigarette was benchmarked against four previously developed epigenetic scores for smoking, as well as a single-site biomarker based on *AHRR* methylation at cg05575921, in wave 1 of LBC1936 ($n = 882$). The results of this comparison can be found in Table 1. mCigarette yielded improved incremental $R^2$ estimates compared to EpiSmokEr, the score developed by McCartney et al.[10] and GrimAge DNAm pack years.

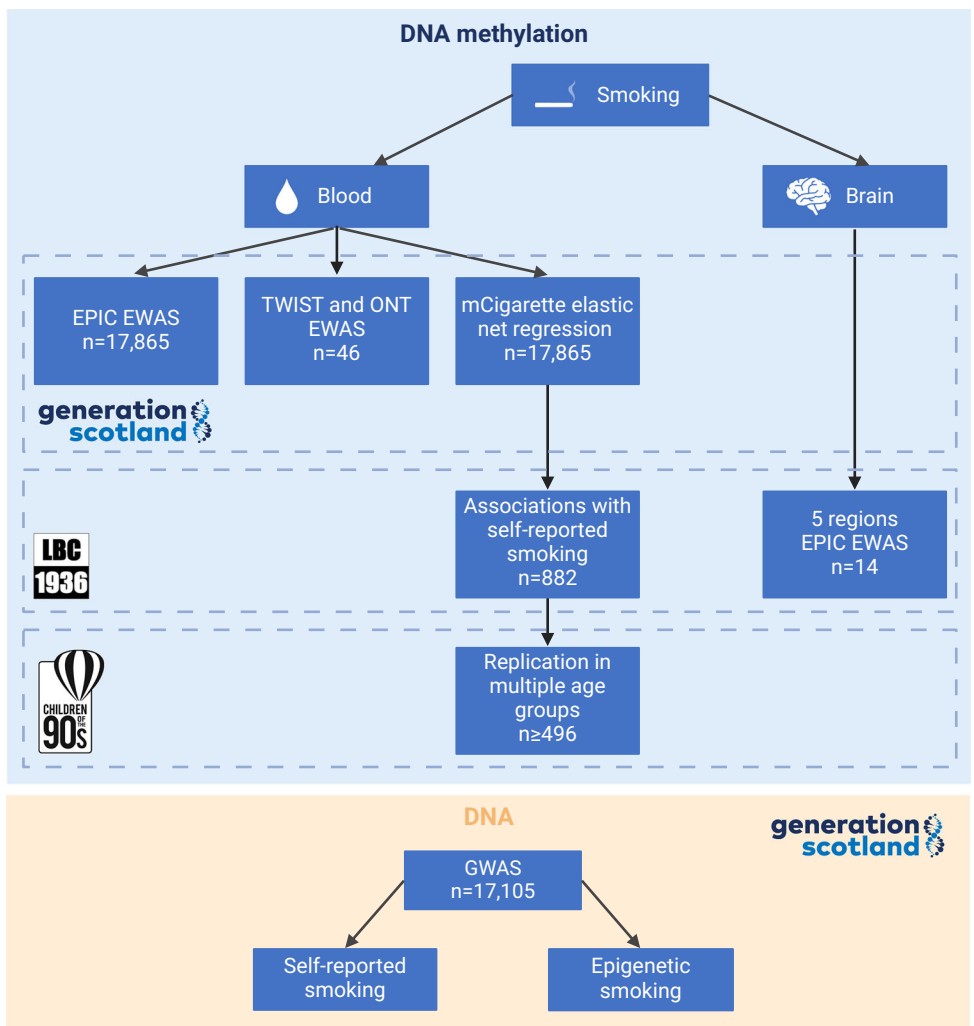

**Fig. 1 | Project overview.** The analysis was conducted using data from three British cohorts: Generation Scotland (GS), the Lothian Birth Cohort of 1936 (LBC1936) and the Avon Longitudinal Study of Parents and Children (ALSPAC). While blood-based DNA methylation (DNAm) data were available in all three cohorts, LBC1936 also contained information about the DNAm levels in post-mortem brain tissue, covering five brain regions from 14 individuals. A Bayesian Epigenome-Wide Association Study (EWAS) of smoking was performed in GS (~850k sites, Illumina EPIC array). For 23 pairs of age- and sex-matched smokers and non-smokers from GS, a high-resolution methylation measurement approach was implemented (~4 million sites, TWIST human methylome panel and ~21 million sites, Oxford Nanopore Technologies sequencing), followed by an EWAS analysis. An epigenetic biomarker of smoking, mCigarette, was developed in GS and tested as a predictor of self-reported smoking in LBC1936. The association between mCigarette and self-reported smoking was replicated in multiple age groups present in ALSPAC. Next, EWASs of smoking were run across five brain regions for 14 individuals using EPIC DNAm from LBC1936. Finally, Genome-Wide Association Studies (GWAS) were run in GS to compare genetic signal of self-reported and epigenetic smoking (GrimAge DNAm pack years score). EPIC – Illumina EPIC array, TWIST – TWIST Biosciences Human Methylome Panel, ONT – Oxford Nanopore Technologies Sequencing. Created in BioRender. Marioni, R. (2024) https://BioRender.com/h44z126.

Subsequently, the weights used to construct the studied scores (apart from the GrimAge DNAm pack years, as weights are not publicly available) were applied to methylation data in the ALSPAC cohort. Within this dataset, no single methylation score consistently outperformed others in distinguishing between current, former, and never smokers across studied age groups. While in young adults (mean age 18 and 24 years old) EpiSmokEr and the score constructed by McCartney et al.[10] achieved the highest AUCs (median AUC = 0.720 [IQR:0.642-0.763]), mCigarette and EpiSmokEr showed excellent performance in older adults (mean age 29 and 50 years old, median AUC = 0.890 [IQR:0.771-0.931]). Full results of the replication study are available in Supplementary Data 8 and are visualised in Supplementary Fig. 2.

**Tissue specificity**

To assess whether the findings translated across different tissue types, the association between tobacco use and DNAm levels across five brain regions were explored using *post-mortem* samples from LBC1936 ($n = 14$, $n_{hippocampus} = 13$). At significance threshold $P < 1 \times 10^{-5}$, five loci in the hippocampus (BA35), one locus in dorsolateral prefrontal cortex (BA46), four loci in primary visual cortex (BA17), nine loci in anterior cingulate cortex (BA24) and three loci in ventral/lateral inferior temporal cortex (BA20/21) were associated with smoking status (Supplementary Data 9). There was no overlap between the significant loci across the studied brain regions.

Some loci demonstrated nearly perfect discrimination of smoking status in blood and brain; however, these loci did not overlap (Fig. 4 and Supplementary Fig. 3). For instance, the methylation status at cg05575921, annotated to the *AHRR* gene, is a well-established marker of smoking status in whole-blood DNAm. However, this marker did not discriminate smoking category in hippocampal DNAm. On the other hand, cg26381592, annotated to the *PMS1* gene, did not effectively distinguish smokers in blood samples, but it exhibited a strong correlation with smoking status in hippocampus samples.

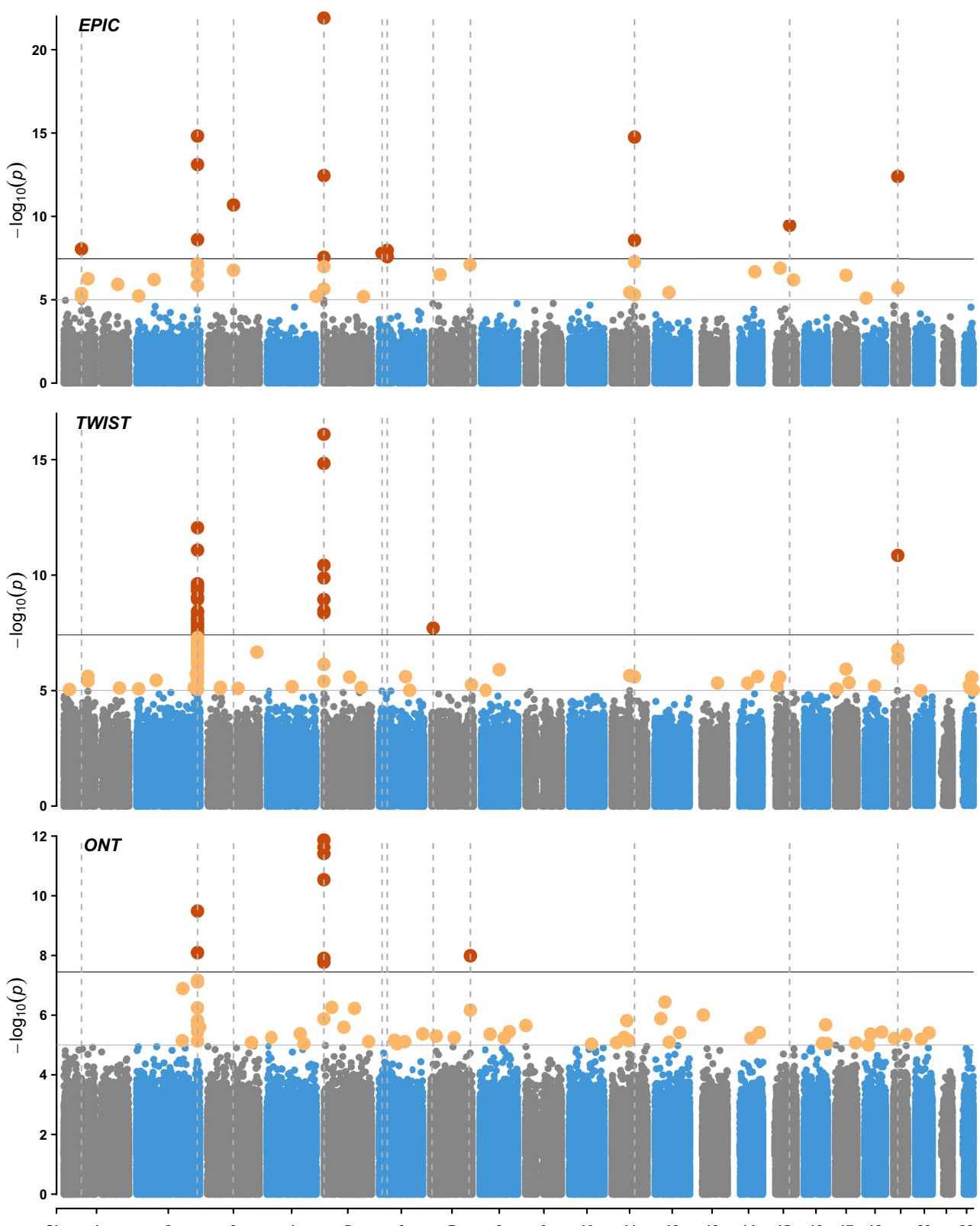

**Fig. 2 | Epigenome-Wide Association Study (EWAS) of current versus never smokers in Generation Scotland (n = 23 pairs).** Analyses were performed using DNA methylation data obtained using the Illumina EPIC array ( ~ 850k CpG sites), the TWIST human methylation panel (4 million CpG sites, targeted short read sequencing) and Oxford Nanopore Technologies sequencing (21 million CpG sites, long read sequencing). The X-axis represents chromosomes 1–22, while the Y-axis shows −log10(P-values). The top horizontal line marks genome-wide significant associations ($P < 3.6 \times 10^{-8}$, red dots), based on the multiple testing threshold estimated by Saffari *et al.*[13]. The bottom horizontal line denotes the suggestive significance threshold ($P < 1 \times 10^{-5}$, yellow dots). Dotted vertical lines highlight loci associated with smoking status at $P < 3.6 \times 10^{-8}$. All statistical tests were two-sided. EPIC – Illumina EPIC array, TWIST – TWIST Biosciences Human Methylome Panel, ONT – Oxford Nanopore Technologies Sequencing.

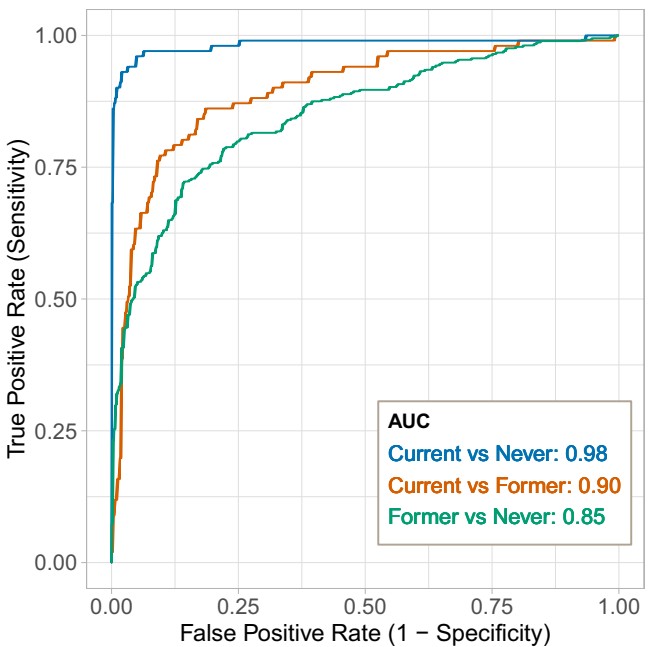

**Fig. 3 | Predictive performance of the epigenetic biomarker of smoking (mCigarette).** The Areas Under the Curves (AUCs) represent the ability of the model to distinguish between the following smoking categories: blue for current smokers vs. never smokers, orange for current smokers vs. former smokers, and green for former smokers vs. never smokers. Source data are provided as a Source Data file.

Subsequently, we tested the performance of a blood-derived DNAm biomarker of smoking (mCigarette) in the brain tissue. When applied to brain DNAm data, mCigarette did not distinguish between smoking categories in any of the studied brain regions (Supplementary Fig. 4).

### GWAS of self-reported and epigenetic smoking
Finally, GWASs of self-reported and epigenetic smoking were conducted. To avoid overfitting, we used the GrimAge DNAm pack years estimator (trained externally in 1731 individuals from the Framingham Heart Cohort study[14]) instead of mCigarette, which was trained in GS.

In 17,105 GS individuals, there was a single nucleotide polymorphism (SNP)-based heritability of 27.3% ($P = 7.0\text{x}10^{-46}$) for self-reported pack years of smoking compared to 41.0% ($P = 8.2\text{x}10^{-98}$) for GrimAge DNAm pack years (Fig. 5). The genomic inflation factors (λ) were 1.06 and 1.10 for pack years and GrimAge DNAm pack years, respectively. At the genome-wide significance level of $P < 5.0\text{x}10^{-8}$, only one locus (rs117836409 annotated to *GDPD1*) was associated with self-reported pack years, in contrast to 39 SNPs (three lead SNPs at two genomic risk loci) associated with GrimAge DNAm pack years (detailed in Supplementary Data 10 and 11). Of the three lead SNPs associated with GrimAge DNAm pack years, two (rs1800440 and rs6495309, annotated to *CYP1B1* and *CHRNA3 - CHRNB4*, respectively) have been previously documented in the GWAS catalogue, which aggregates data from published GWAS studies[15] (Supplementary Data 12). They have been associated with carcinogenesis and nicotine dependence, respectively[16,17]. The SNP not previously annotated in the GWAS Catalog (rs114342890) maps to *RMDN2:RMDN2-AS1*, a long non-coding RNA previously studied in relation to eosinophil counts and melanoma[18,19]. According to GeneHancer, an online database of enhancers, promoters and their inferred targets, all 27 regulatory elements which target *RMDN2:RMDN2-AS1* also regulate the expression of *CYP1B1*[20]. The beta coefficients of the lead loci identified in the GrimAge DNAm pack years GWAS and the smoking pack years GWAS are compared in Supplementary Fig. 5.

The three lead SNPs have been characterised as methylation quantitative trait loci (mQTLs). They all act in *cis* on 43 CpGs (Supplementary Data 13). Only one of these CpGs (cg06264984) is featured in mCigarette, while none are included in EpiSmokEr – the list of CpGs included for the GrimAge DNAm pack years is not publicly available.

Next, the GrimAge DNAm pack years GWAS results were compared to previously published GWAS studies of tobacco use (Supplementary Data 14). At a significance level of $P < 5\text{×}10^{-8}$, seven SNPs annotated to *CHRNA3* and *CHRNA5* aligned with the findings of the largest pack years GWAS to date ($n = 131,892$)[21]. Thirty-seven SNPs, mapping to *CHRNA3, CHRNA5*, and *CHRNB4*, overlapped at $P < 5\text{x}10^{-8}$ with the results of a related phenotype, cigarettes per day ($n = 618,489$)[22].

GrimAge DNAm pack years and self-reported pack years were moderately correlated (Spearman's r = 0.65). The genetic correlation ($r_g$) between GrimAge DNAm pack years and pack years from Erzurumluoglu et al.[21] was 0.62 (SE = 0.12, $P = 4.4\text{x}10^{-7}$), with an LD score regression intercept of 0.99 (SE = 0.01). The $r_g$ between self-reported pack years in the 17,105 GS and pack years from Erzurumluoglu et al.[21] ($n = 131,892$) was 0.67 (SE = 0.19, $P = 5.0\text{x}10^{-4}$), with an LD score regression intercept of 0.99 (SE = 0.01). Additional information on the genetic correlation between epigenetic smoking and previously studied self-reported smoking behaviours (ranging from −0.62 to 0.72)[22] can be found in Supplementary Fig. 6.

## Discussion
This multi-tissue, multi-cohort analysis of the relationship between smoking and DNAm (assessed via arrays and sequencing) has improved both our understanding of the biological consequences of smoking and our ability to measure it objectively. The array-based study, which identified two loci not listed in the EWAS catalog as being associated with smoking, represents the largest single cohort EWAS of smoking and the largest EPIC array EWAS of smoking, to date. The updated epigenetic biomarker of tobacco-use, mCigarette, reliably predicted smoking status and was strongly correlated with self-reported pack years of tobacco use. The analysis of sites differentially methylated in the brains of smokers and non-smokers revealed evidence of tissue-specific signals. There was a partial overlap between the results of the GrimAge DNAm pack years GWAS conducted in GS ($n = 17,105$) and the most extensive GWAS of self-reported smoking to date ($n = 131,892$).

Among the loci not present in the EWAS catalog but identified in the EPIC array EWAS of smoking at PIP > 80%, five array-based CpGs are annotated to *FGF20*, *SCAMP5*, *GRIK5*, *SKI*, and *HOXA4*. *FGF20* plays a key role in the survival and function of dopamine-producing neurons[23], which are crucial in the brain's reward system[24] that nicotine stimulates. *SCAMP5* is essential for dopamine release[25] and the pleasurable sensations associated with smoking. It reinforces smoking behaviour and potentially make individuals more susceptible to nicotine addiction. *GRIK5* encodes a glutamate receptor[26] which modulates dopaminergic neurons within the brain's reward pathways, influencing how strongly these pathways respond to nicotine[27]. A proto-oncogene called *SKI* regulates cell growth and apoptosis[28]. It could potentially modify neural circuitry related to addiction, which affect the risk of developing nicotine dependence or influence the severity of addiction. Lastly, *HOXA4* is a homeobox domain gene that is normally involved in embryonic development[29]. Homeobox genes are abnormally expressed in cancer cells and changes in the expression of *HOXA4* has been specifically associated with colorectal, ovarian and lung cancer[29].

The findings from the next generation sequencing EWASs, which were less well-powered, also underscored the role of smoking in carcinogenesis and disrupted neurodevelopment. Significant loci not listed in the EWAS catalog ($P < 1\text{×}10^{-5}$) identified in the TWIST EWAS included sites mapping to *TSPAN5*, which regulates tumour suppressor gene expression[30]; *USP42*, involved in head and neck cancer pathogenesis[31]; and *SST*, encoding somatostatin, a hormone

**Table 1 | Benchmarking of mCigarette against six biomarkers of smoking: EpiSmokEr[7], GrimAge DNA methylation (DNAm) pack years[14], and three scores developed using Generation Scotland (GS) data: BayesR score[59], a score developed by McCartney et al.[10] and two scores developed in this study – single-site biomarker based on AHRR (cg05575921) blood DNAm level and mCigarette**

| Metric | AHRR | EpiSmokEr | BayesR | McCartney et al.[10] | GrimAge | mCigarette |
|---|---|---|---|---|---|---|
| N training | 17,865 | 1793 | 9448 | 5087 | 1731 | 17,865 |
| a) Variance explained in measured pack years (R²) and correlation (r) metrics | | | | | | |
| Incremental R² | 0.329 | 0.351 | 0.514 | 0.330 | 0.419 | 0.534 |
| r | 0.589 | 0.610 | 0.735 | 0.581 | 0.664 | 0.750 |
| b) Binary classification performance (AUC) | | | | | | |
| Current / Never | 0.972 | 0.985 | 0.977 | 0.982 | 0.983 | 0.984 |
| Current / Former | 0.930 | 0.916 | 0.853 | 0.921 | 0.939 | 0.897 |
| Former / Never | 0.742 | 0.755 | 0.846 | 0.725 | 0.815 | 0.852 |

The BayesR score and the score developed by McCartney et al.[10] were trained on smaller subsets of the GS DNAm dataset. Table a) compares the variance in self-reported pack years explained by null and full models. While the null model was adjusted for age and sex, the full model also included the studied score. The difference between variance explained by null and full models is denoted as the incremental R². Pearson's correlation coefficient is referred to as r. b) Performance of the studied scores in distinguishing between different smoking categories. Binary classification performance of the scores was measured using Areas Under the Curve (AUC).

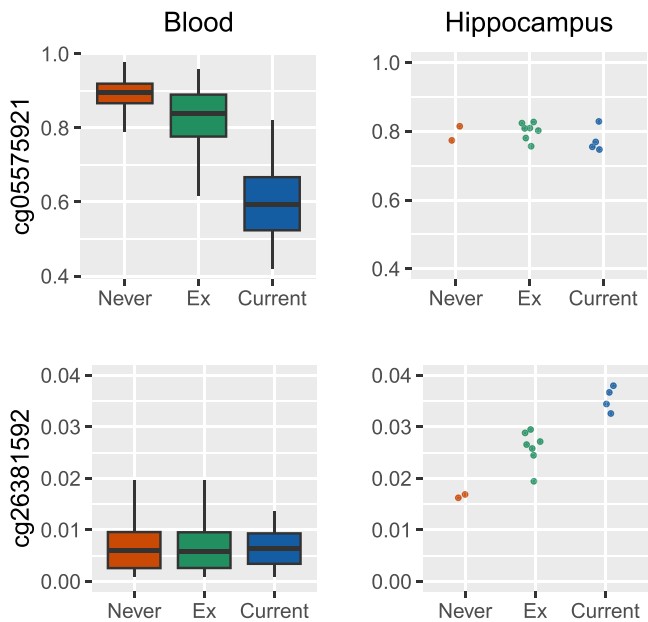

**Fig. 4 | CpG methylation levels in blood** (Lothian Birth Cohort 1936 baseline, $n_{blood\_DNAm}$ = 882) and brain across different self-reported smoking categories (Lothian Birth Cohort 1936, $n_{hippocampus\_DNAm}$ = 13). Box plots are defined as follows: the centre line represents the median (50th percentile). The box bounds indicate the interquartile range (IQR; 25th to 75th percentile). Whiskers extend to the smallest and largest values within 1.5×IQR. Source data are provided as a Source Data file.

implicated in the development of pancreatic cancer[32]. Significant loci revealed by ONT EWAS included CpGs associated with *SEPTIN9*, a tumour suppressor gene[33]; *TERF2*, a telomeric protein linked to tumour formation and progression[34]; and *CNTNAP2*, a potential marker of tumour aggressiveness in oligodendrogliomas[35], which is also implicated in neurodevelopmental disorders. An enrichment analysis of the TWIST results indicated that smoking influences methylation patterns across genes involved in extracellular matrix interactions, chronic inflammation, platelet activation, and ATP regulation, among other pathways. Again, the genes present in enriched pathways typically included *COL4A3* and *COL4A4* which play crucial roles in cancer invasion and metastasis[36]; and inflammation and remodelling of the lung extracellular matrix[37].

When compared to previously published epigenetic biomarkers of tobacco use, mCigarette was a better predictor of smoking pack years. It also showed an excellent performance in discriminating smoking status (current, former, never) in two external cohorts. The robust performance of mCigarette among pregnant women in ALSPAC may reflect the unique biological context of pregnancy, where hormonal changes and epigenetic plasticity make smoking-related epigenetic changes more pronounced. Loci which are most responsive to smoking during pregnancy could provide insights into changes transmitted to the foetus, and affecting the child's health later in life. In the future, mCigarette could be used to monitor smoking cessation efforts during pregnancy, ensuring compliance with cessation programs and potentially improving their effectiveness. The *AHRR* single-site biomarker provided a highly practical option for smoking classification, particularly in settings where limited data points or resources are available. However, multi-site models offered greater precision, accounting for a broader smoking-related methylation signature.

While individual CpG sites offered excellent discrimination of cases and controls, these loci varied by tissue. This is consistent with the low correlations in methylation patterns between blood and brain tissue reported previously[38]. Future work should explore if tissue-specific signals identify pathways and mechanisms by which smoking influences brain health.

In GS, the GrimAge DNAm pack years GWAS results did not align with self-reported smoking GWAS findings but did show partial overlap of lead loci with the most extensive GWAS of self-reported smoking to date. This may suggest an increased power to detect significant loci when the epigenetic score is analysed as a phenotype. However, the genetic correlation between GrimAge DNAm pack years and the meta-analysis smoking pack years was lower than the latter and self-reported pack years in GS. The moderate correlation could reflect differences in the biological pathways that phenotypic and epigenetic measures of smoking are capturing. The GrimAge-based DNAm estimator is designed to capture the cumulative biological impact of smoking, which might include broader aging-related processes beyond direct tobacco exposure. In contrast, the smoking pack years represent a more straightforward measure of cumulative smoking exposure. The shared loci between the GrimAge DNAm pack years GWAS and previous self-reported smoking GWAS included *CHRNA3*, *CHRNA5*, and *CHRNB4*. These genes encode subunits of the nicotinic acetylcholine receptor, responsible for neurotransmission and binding of nicotine in the brain. Variations in these genes can affect nicotine dependence and may be associated with neurological conditions as well as lung cancer[39,40].

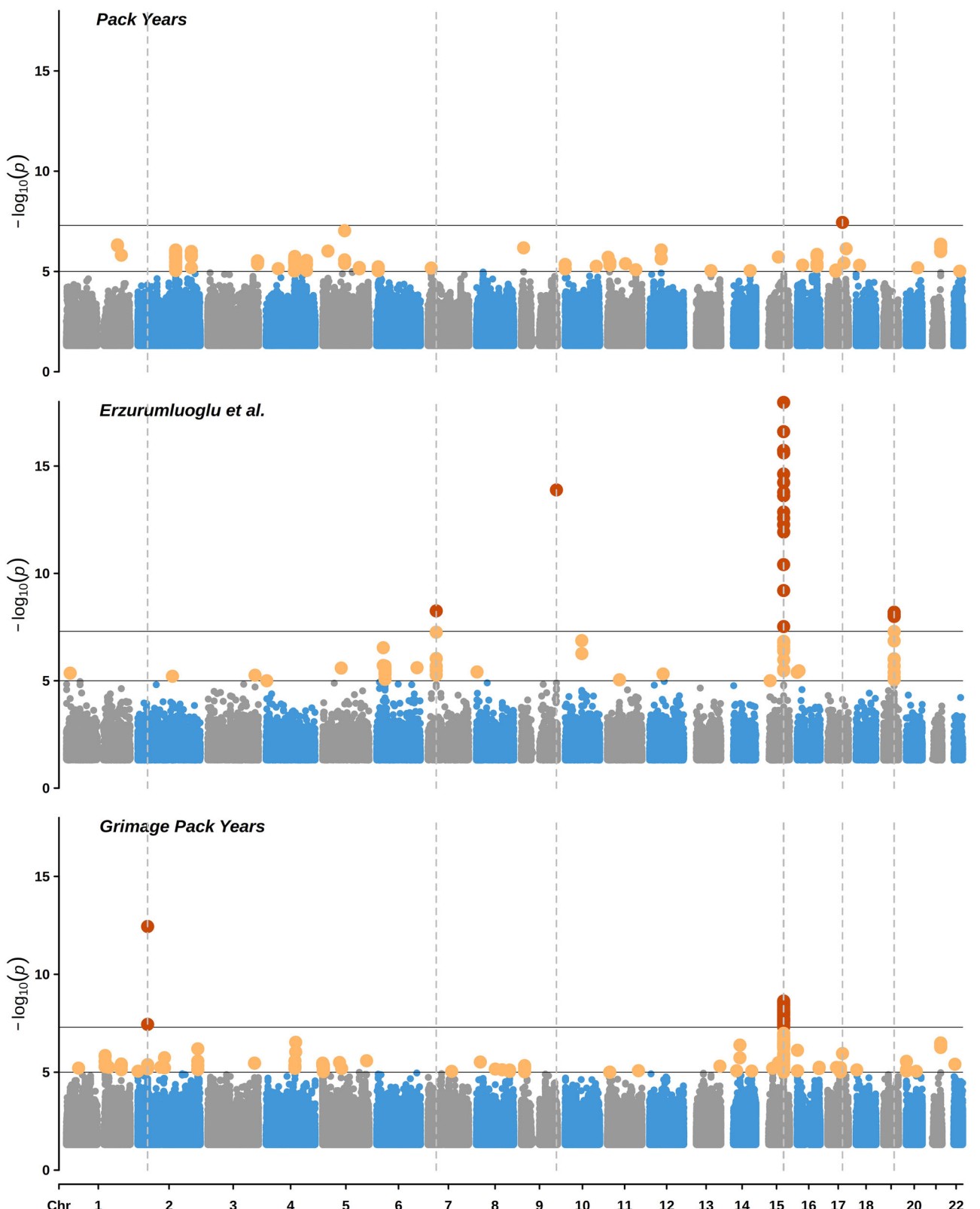

**Fig. 5 | Manhattan plots visualising overlap between various Genome-Wide Association Studies (GWASs) of smoking.** Studied phenotypes included: pack years (Generation Scotland, $n = 17,105$), pack years (largest meta-analysis to date[21], $n = 131,892$), GrimAge-based DNAm estimator of smoking pack years ($n = 17,105$). The X-axis represents genomic positions, while the Y-axis shows −log10(P-values).

The top horizontal line marks genome-wide significant associations ($P < 5×10^{-8}$, red dots), based on the Bonferroni threshold for multiple testing. The bottom horizontal line denotes the suggestive significance threshold ($P < 1×10^{-5}$, yellow dots). Dotted vertical lines mark genomic positions of significant associations at $P < 5×10^{-8}$. All tests were two sided.

The key strengths of this study include the high resolution of epigenetic data used in the main EWAS analysis (both sample size and the number of measured CpG sites), diverse DNAm profiling techniques and the availability of multi-tissue DNAm. Using targeted sequencing allowed for identifying associations between smoking and loci not included on the EPIC chip. Given the mean age of GS volunteers, mCigarette represents many years of exposure to cigarette smoke, and is likely to capture long term DNAm changes associated with smoking.

Limitations of this study include its potential lack of generalizability to non-Europeans and the small number of brain- and whole methylome samples. Nevertheless, given that smoking is associated with major epigenetic alternations, the effects of tobacco use are detectable, even in small datasets. The absence of serum cotinine concentrations prevented us from comparing mCigarette against a clinically established smoking biomarker. We were also unable to verify self-reported smoking status. To mitigate potential bias, we cross-referenced the reported smoking status with other variables (such as self-reported smoking initiation and cessation) and eliminated records with conflicting responses. Additionally, we acknowledge a limitation in our study regarding second-hand smoke exposure. In family-based cohorts such as GS, where related participants may live in the same household, passive smoking is an important factor that could confound the associations between direct smoking and health outcomes. Unfortunately, data on second-hand smoke exposure were not collected in this cohort.

In conclusion, this study explored methylation patterns associated with smoking in blood and brain. The blood-based analyses, using both sequencing- and array-based approaches, identified additional loci associated with tobacco use and led to the development of a highly accurate blood-based DNAm biomarker for smoking. Furthermore, the study provided insights into the differential effects of smoking across tissues. Together, these enhance our understanding of the epigenetic architecture of smoking and shed light on the molecular mechanisms by which tobacco use influences health.

## Methods

### Generation Scotland

Volunteer recruitment to GS has been detailed in a previous publication[41]. Between 2006 and 2011, patients of collaborating general medical practices in Scotland between 35 and 65 years of age were selected at random and invited to take part in the study. These individuals were then encouraged to recruit family members to volunteer to join the cohort. A total of 24,088 individuals between 18-99 years of age completed a health questionnaire. All individuals were asked to report their smoking status (classified as non-smoker, ex-smoker who stopped more than 12 months ago, ex-smoker who stopped within the past 12 months, and current smoker). In addition, current and former smokers provided information about the age they started smoking, the age they quit smoking (former smokers only), and the number of cigarettes they smoked per day. Pack years were computed by multiplying years of smoking by the number of cigarettes smoked per day divided by 20 (number of cigarettes in a pack), and assigning a value of zero to those who never smoked. Further information about the distribution of phenotypic data is available in Supplementary Data 1. DNA extracted from whole blood collected at the baseline visit was genotyped using the Illumina HumanOmniExpressExome array (8v1-2 and 8v1) for 19,992 individuals. Following quality control (QC), imputation to the Haplotype Reference Consortium (HRC) panel[42] and post imputation QC, 7,626,922 SNPs remained for downstream analyses. Methylation levels were measured with Illumina EPIC850k array (18,413 individuals, 752,722 sites after QC). GWAS and EWAS quality control steps have been described before[10] and are also documented in Supplementary Data 15 and 16. Phenotype pre-processing steps are detailed in Supplementary Fig. 7.

### Lothian Birth Cohort 1936

The Lothian Birth Cohort of 1936 ($n = 1091$) comprises community-dwelling older adults in Scotland, most of whom completed an intelligence test aged around 11 years in 1947. Later in life, those living in the Edinburgh and Lothians region were recruited to the cohort at a mean age of ~70 years and then followed up at 3-yearly intervals. Data collected at each wave comprised cognitive test scores as well as biological measures obtained from blood samples. During a baseline interview at age 70, the participants' self-reported smoking status (never smoker, past smoker, current smoker) and smoking behaviour (age at starting, age at stopping, average number of cigarettes smoked per day) were determined. Pack years were calculated as in GS. A brain tissue bank was established at wave 3 (from age ~76 years). Detailed information about the cohort, brain imaging and post-mortem brain samples can be found in a cohort update and brain protocol papers[43,44]. DNAm from whole blood has been measured using Illumina Infinium HumanMethylation450 BeadChip array, while DNAm in five post-mortem brain tissues was profiled using the Illumina EPIC850k array[45]. Quality control and processing details are provided in Supplementary Data 15. Phenotype pre-processing steps are detailed in Supplementary Fig. 8.

### ALSPAC

ALSPAC is a cohort study conducted among pregnant women residing in Avon, UK, with expected delivery dates falling between 1st April 1991 and 31st December 1992[46,47]. Out of the 20,248 eligible pregnancies, 14,541 were enrolled to the study, resulting in 14,062 live births, of which 13,988 children survived to age one. During pregnancy, mothers invited fathers to take part in the study. A total of 12,113 fathers completed questionnaires, with 3807 currently formally enrolled. For a subset of ALSPAC participants (mothers, fathers and children) DNAm was assayed as part of the Accessible Resource for Integrated Epigenomic Studies (ARIES) initiative[48,49]. DNA was extracted from blood samples collected at various time intervals between birth and death, and methylation levels were measured using the Illumina Infinium HumanMethylation450 or MethylationEPIC BeadChip arrays. 450,838 CpG sites passed quality control and were common to these methylation arrays. This study used four collections of DNAm data. Antenatal collection includes data from the ALSPAC mothers only, the Focus on Mothers (FOM)/ Focus on Fathers (FOF) collection corresponds to the mothers/fathers at midlife (~50 years), and the F17 and F24 collections contain ALSPAC children at ages 15–17 (time-point '15up') and 24 (time-point 'F24'), respectively[50]. Smoking status for F17 was based on three questionnaires administered ages 14–17. Former smokers reported having quit at the age 17 questionnaire. Current smokers reported that they had smoked weekly in one or more questionnaires but had not quit. Never smokers reported having never smoked at least one time and never reported having smoked. Smoking status for F24 was assessed using six questionnaires administered ages 14-24. Former smokers reported having smoked regularly at some point prior to age 24, but at age 24 reported not having smoked in the previous 30 days. Current smokers reported at age 24 having smoked in the last 30 days and having smoked at least 50 cigarettes in their lifetime. Never smokers reported never having smoked at age 24. Maternal antenatal smoking was assessed by questionnaires administered at 18- and 32-weeks gestation. Former smokers reported having smoked previously but have stopped smoking for the pregnancy. Current smokers reported smoking regularly in the first trimester. Never smokers reported having never smoked before or during the pregnancy. Smoking status of FOM mothers was assessed using 15 questionnaires administered at study child ages up to age 12 and a final questionnaire at age 18. Former smokers reported having smoked on at least one questionnaire but reported not smoking on the 18 y questionnaire. Current smokers reported being current regular smokers on the 18 y questionnaire. Never smokers consistently reported never having smoked on questionnaires. Smoking status of FOF fathers was

assessed using 11 questionnaires administered at study child ages up to age 12 and a final questionnaire at age 20. Former smokers reported having smoked on at least one questionnaire but reported not smoking on the 20 y questionnaire. Current smokers reported being current regular smokers on the 20 y questionnaire. Never smokers consistently reported never having smoked on questionnaires.

Data for the study were gathered and administered utilizing REDCap electronic data capture tools, which are hosted at the University of Bristol. REDCap (Research Electronic Data Capture) is a secure web application specifically designed to facilitate data capture for research[51]. The study website provides comprehensive details on all available data, accessible through a fully searchable data dictionary located at http://www.bristol.ac.uk/alspac/researchers/our-data/.

### Inclusion & ethics

This study is based on self-reported biological sex (cross-referenced with genetic data). Detailed sex distributions are provided in Supplementary Data 1. Sex/gender was not a primary focus of this study, and analyses were conducted on the full cohort to maximize statistical power. Participants in GS, LBC1936 and ALSPAC did not receive major financial compensation for participation.

All components of GS received ethical approval from the NHS Tayside Committee on Medical Research Ethics (REC Reference Number: 05/S1401/89). All participants provided broad and enduring written informed consent for biomedical research. This study was performed in accordance with the Helsinki declaration.

Ethical approval for the LBC1936 study was obtained from the Multi-Centre Research Ethics Committee for Scotland (MREC/01/0/56) and the Lothian Research Ethics committee (LREC/1998/4/183; LREC/2003/2/29). Use of human tissue for post-mortem studies has been reviewed and approved by the Edinburgh Brain Bank ethics committee and the ACCORD medical research ethics committee, AMREC (ACCORD is the Academic and Clinical Central Office for Research and Development, a joint office of the University of Edinburgh and NHS Lothian). All participants provided written informed consent. These studies were performed in accordance with the Helsinki declaration.

Ethical approval for the ALSPAC study was obtained from the ALSPAC Ethics and Law Committee and the Local Research Ethics Committees. Consent for biological samples has been collected in accordance with the Human Tissue Act (2004). Informed consent for the use of data collected via questionnaires and clinics was obtained from participants following the recommendations of the ALSPAC Ethics and Law Committee at the time. The authors assert that all procedures contributing to this work comply with the ethical standards of the relevant national and institutional committees on human experimentation and with the Helsinki Declaration of 1975, as revised in 2008.

### Sequencing-based approach

Whole methylome sequencing data were generated for 48 unrelated smokers and non-smokers from GS as part of this study. Two approaches were used: the TWIST methylome panel (~4 million CpG sites) and ONT sequencing (~21 million CpG sites). To ensure a robust methylation signal would be present when comparing cases against controls, only heavy smokers (12 males and 12 females chosen from a pool of 40 potential cases with tobacco use ranging from 53.9 to 87.7 pack years; see Supplementary Fig. 9) were selected as cases. Additional details on the sample selection process are provided in the Supplementary Methods. Controls were matched by age and sex using the Matchit package in R[52], with the maximum age difference of less than 12 months (0 years) within a matched pair. Cases and controls showed a clear separation in terms of their methylation at the *AHRR* CpG probe cg05575921 (Supplementary Fig. 10).

Sequencing using the TWIST Human Methylome Panel was performed by the Genetics Core, Edinburgh Clinical Research Facility according to TWIST Targeted Methylation Sequencing Protocol[53].

Sequencing using the ONT kit was performed by Edinburgh Genomics (the first 24 libraries, without basecalling) and the Genetics Core, Edinburgh Clinical Research Facility on the Oxford Nanopore PromethION 24, with R10.4.1 flow cells, running for 72 hours. Further details available in Supplementary Methods. For TWIST pre-processing of raw FASTQ files, read aligning to human reference genome (GRCh38, $n = 29,401,795$ total reference CpGs) with bwa-meth and quality-control of the results was performed using MethylSeq bioinformatics analysis pipeline, version 2.2.0[54,55]. This analysis yielded information about the methylation level and depth of coverage (DoC) at 18,248,472 covered CpG sites.

Dorado, optimized for NVIDIA GPUs, was used for high-accuracy basecalling and modified base detection in raw ONT data. Reads were aligned to GRCh38 human reference genome. Variants were called with epi2me-labs/wf-human-variation nextflow pipeline, version 23.10.1. Methylation level and depth of coverage was measured at 28,989,402 covered CpG sites.

The bedGraph (TWIST) and bedMethyl (ONT) files were subsequently processed in R version 4.3.1[56] using Methrix package[57]. As part of post-processing, loci a) of extremely low (minimal DoC = 2) and high coverage (beyond 0.99 quantile), and b) overlapping with known cytosine to thymine polymorphisms were removed from the methylation dataset. Finally, a coverage filter was applied, retaining only the loci that were covered in at least 40 samples by: a) 10 or more reads (TWIST), b) 5 or more reads (ONT). This left an analysis sample of 3,391,718 and 21,167,712 CpGs for TWIST and ONT data, respectively. CpG sites were annotated with Annotatr R package[58].

### Blood-based DNAm EWAS

A blood-based EWAS of smoking was carried out in 17,865 GS individuals using BayesR[59]. Before running the EWAS, each CpG was corrected for the effects of age, sex, and batch using linear regression (saving the residuals from each model as the new variable). As BayesR implicitly corrects for confounding effects without requiring a full characterization of all covariates, our models were only adjusted for measured variables i.e., estimated white blood cell proportions were not included. However, we conducted a sensitivity analysis by running an EWAS that included adjustments for estimated cell proportions. The results of this sensitivity analysis showed a strong overlap with the primary findings and are presented in Supplementary Data 17. The smoking phenotype (measured in pack years, with a pack defined as 20 cigarettes) was natural log+1 transformed and adjusted for age and sex using linear regression (again, the residuals were saved and used for the downstream analyses). Both the smoking and the CpG variables were scaled to have mean of 0 and variance of 1. The data served as inputs of a Bayesian penalised linear regression model. Four Gaussian priors were specified to model CpGs with varying effect sizes (mixture variances of 0.1%, 1%, 10% and 100%) along with a discrete spike at the origin to model CpGs with no effect. A Gibbs sampler was used to sample over the posterior distribution, conditioning on the input data. A burn-in of 5000 samples was used, after which every fifth sample was retained across 10,000 iterations. A CpG with a posterior inclusion probability of greater than 0.95 was considered as epigenome-wide significant. Previously unpublished associations were identified by searching the literature and the EWAS catalog[12].

### Comparison between TWIST, ONT and EPIC850k

DNAm of 24 pairs of smokers and non-smokers from GS was profiled using the EPIC array, ONT kit, and TWIST platform (see Methods - Sequencing-based approach). During quality control, one pair was identified as mismatched due an age gap exceeding the pre-defined threshold of 12 months. To preserve the integrity of the study design and prevent potential biases in the analysis, this pair was excluded from subsequent analyses. This left an analysis sample of 46 individuals. For each DNAm profiling method, an EWAS of smoking (pack

years) was conducted. The association between DNAm level at each CpG site (outcome) and binary smoking status, age and sex was modelled using linear regression. The results were displayed on a Manhattan plot generated with the CMplot R package[60]. Gene names associated with CpG sites reaching a suggestive significance threshold ($P < 1 \times 10^{-5}$) were extracted and subjected gene set enrichment analysis in Functional Mapping and Annotation (FUMA) GENE2FUNC tool[61], which implements a hypergeometric test. An FDR-adjusted p-value threshold of 0.05 was applied, and a minimum of 2 overlapping genes within each gene set was required.

### Biomarkers of cumulative smoking

An elastic net biomarker of pack years (mCigarette) was trained in 17,865 GS individuals using the glmnet library in R[62]. As part of data pre-processing, CpG sites were filtered to 18,760 loci associated with smoking at False Discovery Rate (FDR) < 0.05 in a previous meta-analysis EWAS ($n = 18,760$) of tobacco use[6]. Alpha was fixed at 0.5 and the lambda value that minimised the mean prediction error was selected via 10-fold cross validation. The selected model assigned non-zero coefficients to 1255 CpGs. A single-site biomarker for smoking, based on methylation at *AHRR* (cg05575921), was also trained in the same subset of GS individuals ($n = 17,865$) using linear regression. Both mCigarette and the single-site biomarker were tested in wave 1 of LBC1936 ($n = 882$, mean age 70 years), while mCigarette alone was tested in ALSPAC ($n = 496–1207$ across four time points). These biomarkers were benchmarked against three epigenetic scores for smoking: EpiSmokEr score[7,63,64], and two scores derived from previous GS analyses on smaller subsets of the dataset - one based on BayesR weights[59], the other via lasso penalised regression by McCartney et al.[10]. In LBC1936, the predictive performance of mCigarette was additionally compared to that of GrimAge DNAm pack years[14]. Pearson's r was calculated to estimate the degree of correlation between self-reported pack years of smoking and the studied scores. The amount of variance in pack years explained by the studied scores was assessed by comparing $R^2$ estimates of null and full models. While the null model was adjusted for age and sex, the full model also included the studied score. Incremental $R^2$ was calculated as the difference between variance explained by null and full models.

The ability of the scores to distinguish between current, former, and never smokers was assessed by AUC. Receiver operating characteristic (ROC) curves were produced using pROC R package[65]. Additional prediction performance metrics such as PRAUC were obtained using MLmetrics R package[66].

### Tissue specificity analyses in LBC1936

DNAm was measured in 5 brain regions (hippocampus - BA35, dorsolateral prefrontal cortex – BA46, primary visual cortex - BA17, anterior cingulate cortex - BA24, ventral/lateral inferior temporal cortex - BA20/21) from post-mortem brain samples of 14 LBC1936 individuals, with one sample missing from hippocampus. Tissue acquisition and processing details are detailed in Stevenson et al.[45]. Using blood-DNAm measured at wave 1 and brain-DNAm data, exploratory EWAS analyses were performed (CpG ~ smoking category). In these analyses, smoking was treated as a continuous variable encoded as 0 = never smoker, 1 = former smoker, 2 = current smoker. Given the small sample size, nominally significant CpG-smoking associations were defined as having $P < 1 \times 10^{-5}$ and were displayed using a ggplot2 boxplot. Due to the same constraints, an enrichment analysis for significant associations was not carried out.

### GWAS of smoking

Associations between genetic variants and smoking were examined using GWASs. Two phenotypes were considered: natural log(-transformed pack years of smoking + 1) and an epigenetic score for smoking pack years generated by an online calculator that uses the

algorithm derived for the GrimAge epigenetic clock[14]. After initial filtering (see Supplementary Data 3), there were 17,105 GS individuals with data available at 7,626,922 SNPs. Both GWASs were conducted using the GCTA software[67], with a Genetic Relationship Matrix fitted into a fastGWA-lmm model to account for relatedness. Each trait was adjusted for age, sex, and 20 genetic principal components. The results of these analyses, along with the findings of previous GWASs of tobacco use, were visualised using CMplot[60]. Lead and methylation quantitative loci among the results of GrimAge DNAm pack years GWAS were identified using the default settings in FUMA[61] and goDMC[68], respectively. GWAS catalog was accessed via FUMA website. Genetic correlations were calculated with LDSC[69].

### Data availability

Cohort data are available under restricted access. According to the terms of consent for Generation Scotland participants, access to data must be reviewed by the Generation Scotland Access Committee. Applications should be made to genscot@ed.ac.uk and normally take up to six weeks for approval. Further details can be found at https://genscot.ed.ac.uk/ for-researchers/access/. Lothian Birth Cohort data are available on request from the Lothian Birth Cohort Study, University of Edinburgh. Information on data access can be found at https://lothian-birth-cohorts. ed.ac.uk/data-access-collaboration, including data dictionaries and a Data Request Form (DRF). Data access requests, including a completed DRF, should be sent to the study director (simon.cox@ed.ac.uk); requests are normally reviewed by the team within four weeks and data are shared subject to completion of a Data or Material Transfer Agreement. ALSPAC is run as a resource for the research community. Instructions for accessing ALSPAC data can be found here: https://www. bristol.ac.uk/alspac/researchers/access/. A research proposal must be submitted via the research proposal system for consideration by the ALSPAC Executive Committee. For any questions regarding accessing data or samples please email alspac-data@bristol.ac.uk (data) or bbl-info@bristol.ac.uk (samples). Approval may take up to two weeks. The raw data underlying figures are provided in the Supplementary Information/Source Data file. The GWAS and EWAS summary statistic output is available in the Zenodo database [https://doi.org/10.5281/zenodo. 14878399]. For any further correspondence and material requests please contact Dr Riccardo Marioni at riccardo.marioni@ed.ac.uk. Source data are provided with this paper.

### Code availability

All custom R (version 4.3.1), Python (version 3.9.7), and bash code is available with open access at the following Zenodo repository: https:// doi.org/10.5281/zenodo.14882848[70].

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

## Acknowledgements

We sincerely appreciate all Generation Scotland study participants, staff, and research team members for their past and ongoing contributions to these studies. Generation Scotland received core support from the Chief Scientist Office of the Scottish Government Health Directorates (CZD/16/6) and the Scottish Funding Council (HR03006). Genotyping and DNA methylation profiling of the Generation Scotland samples was carried out by the Genetics Core Laboratory at the Edinburgh Clinical Research Facility, Edinburgh, Scotland and was funded by the Medical Research Council UK and the Wellcome Trust (Wellcome Trust Strategic Award STratifying Resilience and Depression Longitudinally (STRADL; Reference 104036/Z/14/Z). The DNA methylation data assayed for Generation Scotland was partially funded by a 2018 NARSAD Young Investigator Grant from the Brain & Behavior Research Foundation (Ref: 27404; awardee: Dr David M Howard) and by a JMAS SIM fellowship from the Royal College of Physicians of Edinburgh (Awardee: Dr Heather C Whalley). The authors thank all LBC1936 study participants and research team members who have contributed, and continue to contribute, to ongoing studies. The LBC1936 is supported by the BBSRC, and the Economic and Social Research Council [BB/W008793/1] (which supports S.E.H.), Age UK (Disconnected Mind project), the Milton Damerel Trust, the Medical Research Council (MR/M01311/1), and the University of Edinburgh. Methylation typing of LBC1936 was supported by the Centre for Cognitive Ageing and Cognitive Epidemiology (Pilot Fund award), Age UK, The Wellcome Trust Institutional Strategic Support Fund, The University of Edinburgh, and The University of Queensland. Genotyping was funded by the BBSRC (BB/F019394/1). S.R.C. is supported by a Sir Henry Dale Fellowship jointly funded by the Wellcome Trust and the Royal Society (Grant Number 221890/Z/20/Z). We are extremely grateful to all the families who took part in this study, the midwives for their help in recruiting them, and the whole ALSPAC team. The UK Medical Research Council and Wellcome (Grant ref: 217065/Z/19/Z) and the University of Bristol provide core support for ALSPAC. This publication is the work of the authors and they will serve as guarantors for the contents of this paper. A comprehensive list of grants funding is available on the ALSPAC website (http://www.bristol.ac.uk/alspac/external/documents/grant-acknowledgements.pdf). Funding for ALSPAC DNAm measurements were supported by the Wellcome (102215/2/13/2); the University of Bristol; the UK Economic and Social Research Council (ES/N000498/1); the UK Medical Research Council (MC_UU_12013/1, MC_UU_12013/2); the Biotechnology and Biological Sciences Research Council (BBI025751/1 and BB/I025263/1); and the John Templeton Foundation (60828). P.Y. and M.S. work is supported by the National Institute for Health and Care Research Bristol Biomedical Research Centre, the Medical Research Council Integrative Epidemiology Unit at the University of Bristol (MC_UU_00032/3, MC_UU_00032/4, MC_UU_00032/6), and Cancer Research UK [C18281/A29019, EDDISA-Jan22\100003]. A.D.C. is supported by a Medical Research Council PhD Studentship in Precision Medicine with funding from the Medical Research Council Doctoral Training Program and the University of Edinburgh College of Medicine and Veterinary Medicine. R.F.H is supported by an MRC IEU Fellowship. E.B. and R.E.M. are supported by Alzheimer's Society major project grant AS-PG-19b-010. This research was funded in whole, or in part, by the Wellcome Trust (104036/Z/14/Z, 108890/Z/15/Z, 220857/Z/20/Z, and 221890/Z/20/Z). For the purpose of open access, the author has applied a CC BY public copyright license to any Author Accepted Manuscript version arising from this submission.

## Author contributions

A.D.C. analysed the data. E.B. and R.F.H. developed the Bayesian EWAS pipeline. P.Y. and M.S. replicated results in the ALSPAC cohort. D.L.M., R.F.H., R.C., L.McG., L.M., S.E.H., J.C., A.C., T.L.S., S.R.C., and K.L.E. were involved in the data generation. A.D.C. and R.E.M. drafted the initial manuscript. A.D.C., J.F.P., K.L.E., and R.E.M. designed the study. All authors read and approved the final manuscript.

## Competing interests

R.E.M. is an advisor to the Epigenetic Clock Development Foundation. R.F.H. has received consultant fees from Illumina. R.E.M. and R.F.H. have received consultant fees from Optima partners. L.M. received speaker fees from Illumina and Oxford Nanopore Technologies. All other authors declare no competing interests.
