## [Transparent Peer Review file · Nature Communications]

Blood- and brain-based EWAS of smoking

Corresponding Author: Ms Aleksandra Chybowska

Version 0:

Reviewer comments:

Reviewer #1

(Remarks to the Author)

(Remarks on code availability)

Reviewer #2

(Remarks to the Author)

This study utilized advanced epigenetic techniques across multiple cohorts to enhance understanding of smoking-related DNA methylation patterns in blood and brain tissues. The authors developed a blood-based biomarker (mCigarette) for smoking status, and highlighted tissue-specific differences in methylation patterns, underscoring the complex biological consequences of tobacco use. These findings contribute to our ability to objectively measure smoking exposure and deepen insights into its molecular impacts on health. My detailed review, encompassing comments and suggestions, is outlined below.

1. In the cohorts of Generation Scotland and Lothian Birth Cohort 1936 (LBC1936), What were the definitions of never, former, and current smoking? Were they consistent with ALSPAC?
2. For the 48 unrelated smokers and non-smokers from GS, can the authors be more specific on the methods used to match the cases and controls? What was the maximum difference in age in the matching?
3. Can the authors discuss the potential passive smoking / secondhand smoke exposure for participants that are related and possibility in the same household within the family-based cohorts studied?
4. The current Figure 2 does not fully present the comparison of the EWAS results based on the EPIC array and TWIST panel. For the CpG sites available in both platforms, an additional scatter plot coupled with a regression slope for contrasting the beta estimates from two platforms would be very helpful. Inflation factors of EWAS results should be reported.
5. How was the batch effect handled in the EWAS? Was it adjusted in the linear regression models as categorical variables to calculate the residuals of pack years? If the cell type proportions were not available, is it possible to estimate (PMID: 22568884)?
6. Are X-chromosome data available for the EPIC/TWIST EWAS? If yes, could the authors consider including the analysis results?
7. How does the blood-based EWAS result under the Bayesian framework compare with the previous studies that used Illumina 450k BeadChip array (N=15,907, PMID: 27651444) and Illumina EPIC 850k array (N=15,014, PMID: 38199042)? Both these two previous investigations used frequentist-based methods. A search of PIP in the EPIC EWAS results for the previously known CpGs sites would be very helpful as a replication.
8. For the EPIC EWAS, the authors performed a comprehensive search in the EWAS Catalog – can the authors expand the discussion of the novel genes identified, such as SCAMP5, FGF20, and SKI? SCAMP5 and FGF20 are involved in neurodevelopment pathways and may affect the brain's response to nicotine and influence addictive behaviors.
9. The TWIST EWAS is particularly interesting due to its high coverage of CpG sites. Given the small proportion of overlap compared with EPIC, most CpG sites are novel. A pathway enrichment analysis would be of interest as it may reveal novel mechanisms in addition to existing understanding.
10. For the training of mCigarette, can the authors explain the reason of choosing elastic net over other machine learning methods such as LASSO or ridge regression? A total of 17865 CpG sites were included out of the total 18760 identified in

Joehanes et al. (PMID: 27651444), with 895 removed from the algorithm. How many of the CpG sites in EpiSmokEr, BayesR and McCartney et al (GrimAge algorithm not available) overlap with mCigarette? Identifying overlap helps in understanding the consistency and robustness of CpG sites associated with smoking across different studies and methodologies. Non-overlapping sites could suggest additional information or potentially new markers captured by mCigarette, which might contribute to enhancing predictive performance or understanding the biological mechanisms further.

11. The authors stated that the performance of mCigarette in prediction among adults with mean age 29 and 50 years was excellent, which may not fully capture the findings across age groups. In Supplementary Table 7, which can be converted into a color-coded bar plot for better illustration, we can see that the performance in the antenatal group (age 29) is the best compared with all other age groups. The antenatal group is 100% female with data collected at 18 and 32 weeks gestation, with only 98 current smokers out of 968 participants. This is an important finding as it may suggest that the epigenetic markers used in the model are highly sensitive to smoking-related changes during pregnancy. Can the authors add this to the discussion section and explore the potential implications?

12. For the DNAm analysis in the 5 brain regions among LBC1936, did the authors consider a sensitivity analysis treating smoking as a continuous variable (pack years)? I wonder if a 1df test would be better powered compared to a 2df test using a categorical variable of smoking status.

13. The inflation factor and the LD Score regression intercept for the GWAS results should be reported, as relatedness was addressed in the statistical model in the family-based cohort. The overlap of lead loci compared between the GrimAge DNAm pack years GWAS and smoking pack years GWAS can be visualized in scatterplot of the beta coefficients in addition to the current Figure 5. What's the correlation between smoking pack years and GrimAge-based DNAm estimator of smoking pack years in GS? Can the authors discuss the potential reasons for the relatively moderate genetic correlation of 0.51 between the two phenotypes?

14. In Supplemental Tables 9-13, the Genome Reference Consortium Human Build version should be specified. Seems like the usage of GRCh37 and GRCh38 is not consistent. Could this help address the missingness of SNPs while comparing across GWAS summaries (Erzurumluoglu et al. was GRCh37, Saunders et al., 2022 was GRCh38)? By briefly searching the SNPs from the Erzurumluoglu et al. summary (https://ftp.ebi.ac.uk/pub/databases/gwas/summary_statistics/GCST007001-GCST008000/GCST007601/), some SNPs labeled as missing are available.

Minor issues:

Page 10, Line 15, first appearance of "FOM/FOF" should be specified.

Supplemental Table 11, the column names should be explained in the footnote. The "Ptext" column can be deleted if all values are not available.

Supplemental Table 12, the column name "CisTrans" and its coding should be added to the footnote.

(Remarks on code availability)

Reviewer #3

(Remarks to the Author)

Chybowska et al. used several approaches to characterize smoking-associated DNA methylation changes in humans. In the Generation Scotland Cohort, Illumina EPIC850K arrays (n=17,865) and TWIST methylome panels (n=48) were used to conduct epigenome-wide association studies (EWASs) on DNA methylation in whole blood. The Generation Scotland data were also used to derive a composite biomarker of cigarette smoke exposure (mCigarette), which was then evaluated against other composite biomarkers of cigarette exposure using DNA methylation data from two independent cohorts (Lothian Birth Cohort of 1936 and ALSPAC). DNA methylation data from whole blood and 5 brain regions of 14 individuals from the Lothian Birth Cohort of 1936 were used to examine tissue-specific effects. In my opinion, the size of the cohort used in the EWAS analysis and in the development of a biomarker of smoking, along with the inclusion of two independent cohorts to evaluate use of mCigarette, contribute to further the understanding of the association between cigarette smoking and DNA methylation in humans.

Major Comments:

1. The methods are somewhat unclear as to which DNA methylation datasets represent newly generated data versus reanalysis of previously published datasets. Although this information is provided in the Reporting Summary, authors should update methods to make this distinction more obvious in the text.

2. Cell composition of whole blood has been demonstrated to be affected by a variety of factors including age, smoking status, and immunological history (Bergstedt et al. 2022). In section 2.5 of Methods (pg. 13, lines 4-6) the authors indicate that EWAS models were not adjusted for estimated white blood cell proportions as "BayesR implicitly corrects for confounding effects." Given the 1) substantial impact that variables not included in this study (e.g., cytomegalovirus serostatus) have on DNA methylation and composition of immune cells in blood and 2) the cell-type specific, smoking-associated DNA methylation changes reported in (Wang et al. 2023), I recommend comparing the DNA methylation EWAS with adjustment for estimated blood cell proportions to results presented in section 3.1 of Results.

3. In the abstract, the authors acknowledge that methylation levels at several CpG sites show "near-perfect discrimination of smoking status" in blood (pg. 3, line 7). Whereas they describe the ability of the mCigarette biomarker to discriminate across smoking categories as "excellent" (pg. 3, line 4). I think it is crucial to benchmark any composite DNA methylation biomarkers of smoking against CpG sites that have been reproducibly demonstrated to discriminate amongst categories of smoking status in multiple cohort studies, specifically DNA methylation at cg05575921 (AHRR) in blood samples. As such, it would be beneficial to include a comparison using cg05575921 as a biomarker of smoking in the evaluation of mCigarette (Results 3.3., Table 1).

Minor Comments:

1. Pg. 3, line 7: Change "both" to "either"
2. Pg. 5, line 3: Change "molecule" to "molecules"
3. Pg. 9, lines 20-22: This sentence indicates the whole blood and brain tissue samples were measured with "HumanMethylation450 BeadChip," whereas, Figure 1 (pg. 7) indicates "EPIC EWAS" and reference to the DNA methylation data cited in line 22 (Stevenson et al. 2022) indicates that Illumina 850K arrays were used to quantify DNA methylation in brain samples and Illumina 450K arrays were used to profile the blood samples for the Lothian Birth Cohort.
4. Pg. 14, line 7-8: This sentence indicates that results from TWIST and EPIC850k arrays were "compared on a Manhattan plot." Please change to "displayed on Manhattan plots" as the separate Manhattan plots in Figure 2. (page 20) do not compare the results of the two methods; they merely display the results from each analysis near each other.
5. Pg. 19, line 5-7: Were all 44 associations from the EPIC-based analysis evaluated in the TWIST-based analysis? That is, were they just not significant at $P < 1 \times 10^{-5}$ in the TWIST analysis or were they excluded for technical reasons (e.g., not passing QC)? Clarification on this point in the manuscript text is recommended.
6. Pg. 19, line 9: Change "presents a comparison of" to "displays."
7. Pg. 19, line 12-13: In this sentence, I think it would be informative to include the number of associations that had previously been reported and the number of loci that were novel (as opposed to stating that some had been reported and some were novel).
8. Pg. 23, line 1: Delete "in"
9. Pg. 25, line 12: Should be cg05575921
10. Pg. 25, line 16: Change "brain tissue samples" to "hippocampus samples" unless this finding (cg26381592) can be generalized to other brain regions tested.
11. Pg. 26, Figure 4: It would be informative to include figures showing DNA methylation at cg05575921 and cg26381592 in the last blood of the individuals that are shown in the hippocampus data.
12. Pg. 31, lines 20-21: Include reference for association between genes and neurological conditions and/or lung cancer.

References:

Bergstedt J, Azzou SAK, Tsuo K, Jaquaniello A, Urrutia A, Rotival M, Lin DTS, MacIsaac JL, Kobor MS, Albert ML, Duffy D, Patin E, Quintana-Murci L; Milieu Intérieur Consortium. The immune factors driving DNA methylation variation in human blood. *Nat Commun.* 2022 Oct 6;13(1):5895. doi: 10.1038/s41467-022-33511-6. PMID: 36202838; PMCID: PMC9537159.

Wang X, Campbell MR, Cho HY, Pittman GS, Martos SN, Bell DA. Epigenomic profiling of isolated blood cell types reveals highly specific B cell smoking signatures and links to disease risk. *Clin Epigenetics.* 2023 May 25;15(1):90. doi: 10.1186/s13148-023-01507-8. PMID: 37231515; PMCID: PMC10211291.

(Remarks on code availability)

Code was well-organized and easy to follow. I did not install and run the code to evaluate reproducibility. README file provided thorough instructions and appropriate references to code developed by other groups and used in these analyses.

Version 1:

Reviewer comments:

Reviewer #1

(Remarks to the Author)

The authors have addressed all concerns raised by reviewers.

(Remarks on code availability)

Reviewer #2

(Remarks to the Author)

(Remarks on code availability)

Reviewer #3

(Remarks to the Author)

Authors have thoughtfully addressed all major and minor comments from my review and included additional analyses and in-text modifications to provide clarification where needed.

(Remarks on code availability)

RESPONSE TO REVIEWER COMMENTS

Reviewer #2 (Remarks to the Author):

This study utilized advanced epigenetic techniques across multiple cohorts to enhance understanding of smoking-related DNA methylation patterns in blood and brain tissues. The authors developed a blood-based biomarker (mCigarette) for smoking status, and highlighted tissue-specific differences in methylation patterns, underscoring the complex biological consequences of tobacco use. These findings contribute to our ability to objectively measure smoking exposure and deepen insights into its molecular impacts on health. My detailed review, encompassing comments and suggestions, is outlined below.

Response: We would like to thank Reviewer #2 for their thoughtful and comprehensive review of our study. We appreciate the recognition of our efforts to improve the understanding of smoking-related DNA methylation changes using advanced epigenetic techniques. We have added our responses to comments below, in blue.

1. In the cohorts of Generation Scotland and Lothian Birth Cohort 1936 (LBC1936), What were the definitions of never, former, and current smoking? Were they consistent with ALSPAC?

Generation Scotland: Smoking status was self-reported and classified as non-smoker, ex-smoker who stopped more than 12 months ago, ex-smoker who stopped within the past 12 months, and current smoker. This definition includes a specific distinction between recent ex-smokers (those who quit within the past 12 months) and those who quit earlier.

We have added a clarification to the definition of former smoking in Generation Scotland.

Page 19, line 9: “All individuals were asked to report their smoking status (classified as non-smoker, ex-smoker who stopped more than 12 months ago, ex-smoker who stopped within the past 12 months, and current smoker)”

LBC1936: Smoking status was self-reported and classified as non-smoker, ex-smoker, and current smoker. Unlike Generation Scotland, LBC1936 did not differentiate between recent and long-term ex-smokers; both are combined under the category of ex-smoker. As this information was already available in the main manuscript text, we did not make any alternations.

ALSPAC: Smoking status was determined differently for each of the cohorts and involved multiple questionnaires across different age ranges.

F17 Cohort: Smoking status was based on responses from three questionnaires administered between ages 14 and 17. Former smokers reported quitting by age 17, current smokers had smoked weekly but not quit, and never smokers consistently reported never smoking.

F24 Cohort: Smoking status was assessed using six questionnaires administered between ages 14 and 24. Former smokers had smoked regularly but not in the past 30 days by age 24, current smokers reported smoking in the past 30 days and at least 50 cigarettes in their lifetime, and never smokers consistently reported never smoking.

Antenatal Smoking: Maternal antenatal smoking was assessed by questionnaires administered at 18 and 32 weeks gestation. Former smokers reported having smoked previously but have stopped

smoking for the pregnancy. Current smokers reported smoking regularly in the first trimester. Never smokers reported having never smoked before or during the pregnancy

Maternal and Paternal Smoking: For both mothers (FOM) and fathers (FOF), smoking status was tracked over time using multiple questionnaires. Former smokers were those who reported smoking at least once but had quit by the final questionnaire, current smokers continued to report smoking regularly, and never smokers consistently reported never smoking.

As this information was already available in the main manuscript text, we did not make any alternations.

In summary, the key differences lie in how each study distinguishes between former and current smokers, the time frames used to define former smoking, and the use of multiple questionnaires over time in ALSPAC to capture smoking status, in contrast to the single-time-point assessments in Generation Scotland and LBC1936.

2. For the 48 unrelated smokers and non-smokers from GS, can the authors be more specific on the methods used to match the cases and controls? What was the maximum difference in age in the matching?

We are grateful for this comment. We have also made the following change to the Methods section:

Page 23, line 21: “To ensure a robust methylation signal would be present when comparing cases against controls, only heavy smokers (12 males and 12 females chosen from a pool of 40 potential cases with tobacco use ranging from 53.9 to 87.7 pack years) were selected as cases. Additional details on the sample selection process are provided in the **Supplementary Methods**. Controls were matched by age and sex using the Matchit package in R⁵², with the maximum age difference of less than 12 months (0 years) within a matched pair..”

3. Can the authors discuss the potential passive smoking / secondhand smoke exposure for participants that are related and possibility in the same household within the family-based cohorts studied?

In family-based cohorts, particularly those involving related participants who may reside in the same household, the potential for passive smoking or second-hand smoke exposure is an important factor to consider. This exposure could have significant implications for studies investigating the health effects of smoking, as it introduces a variable that may confound the association between direct smoking and the studied outcomes. Unfortunately, this information was not collected in Generation Scotland. We have noted this in the limitations section:

Page 17, line 20: “Additionally, we acknowledge a limitation in our study regarding second-hand smoke exposure. In family-based cohorts such as GS, where related participants may live in the same household, passive smoking is an important factor that could confound the associations between direct smoking and health outcomes. Unfortunately, data on second-hand smoke exposure were not collected in this cohort.”

4. The current Figure 2 does not fully present the comparison of the EWAS results based on the EPIC array and TWIST panel. For the CpG sites available in both platforms, an additional scatter plot coupled with a regression slope for contrasting the beta estimates from two platforms would be very helpful. Inflation factors of EWAS results should be reported.

We appreciate Reviewer's #2 suggestion to enhance the presentation of the EWAS results.

We have extended this section by adding an Oxford Nanopore Technologies-based EWAS data (n=46) to Abstract, Introduction, Methods, Results and Discussion.

We have created additional scatter plots comparing the beta estimates for the CpG sites available on the studied platforms, along with the corresponding regression slopes (**Supplementary Fig. 1**). We have also calculated the inflation factor and incorporated it into the text associated with **Figure 2**. The following changes were introduced:

Page 7, line 3: "Next, we extended this analysis by running high resolution EWASs of smoking on a subset of 23 pairs (n=46) of current vs never smokers with Illumina EPIC array (~850k CpG sites), TWIST human methylation panel (~4 million CpG sites), and Oxford Nanopore Technologies (ONT) sequencing data (~21 million CpG sites). At $P < 3.6 \times 10^{-8}$, the EPIC-based analysis revealed 15 CpG sites associated with smoking status (EWAS inflation factor, $\lambda = 0.94$), while the TWIST-based and ONT-based analyses identified 33 ($\lambda = 1.60$) and 9 ($\lambda = 1.11$) associations, respectively. At a less stringent threshold ($P < 1 \times 10^{-5}$), these counts increased to 42, 102, and 63 for the EPIC-, TWIST-, and ONT-based analyses, respectively. The overlap between the sites identified by these technologies is detailed in **Supplementary Data 3. Figure 2** and **Supplementary Fig. 1** (comparison of beta estimates) display the results obtained from the TWIST, ONT and EPIC EWAS.

Among the 33 associations identified as significant in the TWIST EWAS at $P < 3.6 \times 10^{-8}$, two had been previously reported in the EWAS catalog (based on DNAm profiled with array technologies). These included *AHRR* (chr5-373263-373264, beta=-0.35, $P = 1.2 \times 10^{-10}$) and an intergenic locus found on chromosome 2 (chr2-232419951-232419952, beta=-0.24, $P = 3.5 \times 10^{-8}$). The remaining 31 loci significant at this threshold were novel, including sites annotated to *F2RL3* (chr19-16889741-16889742, beta=-0.31, $P = 1.3 \times 10^{-11}$) and *USP42* (chr7-6126706-6126707, beta=0.05, $P = 1.8 \times 10^{-8}$). At a less stringent threshold of $P < 1 \times 10^{-5}$, 98 sites were identified as novel, including *SST* (chr3-187670342-187670343, beta=-0.09, $P = 2.2 \times 10^{-7}$) and *TSPAN5* (chr4-98472405-98472406, beta=-0.06, $P = 6.8 \times 10^{-6}$). Further details are provided in **Supplementary Data 4**."

Other changes:

Abstract: "In this work, we conducted a Bayesian Epigenome-Wide Association Study of smoking pack years (n=17,865, ~850k sites, Illumina EPIC array) and extended it by analysing whole genome data of smokers and non-smokers from Generation Scotland (n=46, ~4–21 million sites via TWIST and Oxford Nanopore sequencing)."

Methods:

Page 24, line 4: "Sequencing using the TWIST Human Methylome Panel was performed by the Genetics Core, Edinburgh Clinical Research Facility according to TWIST Targeted Methylation Sequencing Protocol⁵³. Sequencing using the ONT kit was performed by Edinburgh Genomics (the first 24 libraries, without basecalling) and the Genetics Core, Edinburgh Clinical Research Facility on the Oxford Nanopore PromethION 24, with R10.4.1 flow cells, running for 72 hours. Further details available in **Supplementary Methods**."

Dorado, optimized for NVIDIA GPUs, was used for high-accuracy basecalling and modified base detection in raw ONT data. Reads were aligned to GRCh38 human reference genome. Variants were called with epi2me-labs/wf-human-variation nextflow pipeline, version 23.10.1. Methylation level and depth of coverage was measured at 28,989,402 covered CpG sites.

The bedGraph (TWIST) and bedMethyl (ONT) files were subsequently processed in R version 4.3.1⁵⁶ using Methrix package⁵⁷. As part of post-processing, loci a) of extremely low (minimal DoC = 2) and high coverage (beyond 0.99 quantile), and b) overlapping with known cytosine to thymine polymorphisms were removed from the methylation dataset. Finally, a coverage filter was applied, retaining only the loci that were covered in at least 40 samples by: a) 10 or more reads (TWIST), b) 5 or more reads (ONT). This left an analysis sample of 3,391,718 and 21,167,712 CpGs for TWIST and ONT data, respectively. CpG sites were annotated with Annotatr R package⁵⁸. “

Page 26, line 6: “DNAm of 24 pairs of smokers and non-smokers from GS was profiled using the EPIC array, ONT kit, and TWIST platform (see Methods - Sequencing-based approach). During quality control, one pair was identified as mismatched due an age gap exceeding the pre-defined threshold of 12 months. To preserve the integrity of the study design and prevent potential biases in the analysis, this pair was excluded from subsequent analyses. This left an analysis sample of 46 individuals.”

Results, Page 8, line 1: “In the ONT EWAS, 9 sites were significant at $P < 3.6 \times 10^{-8}$, of which only one had been previously catalogued: a site mapping to *AHRR* (chr5-373263-373264, $\beta = -0.47$, $P = 2.4 \times 10^{-12}$). The remaining eight were novel, including additional loci within the *AHRR* region, loci from an intergenic region on chromosome 2 (e.g., chr2-232420079-232420080, $\beta = -0.37$, $P = 3.4 \times 10^{-10}$) and a site annotated to *CNTNAP2* (chr7-147245588-147245589, $\beta = 0.37$, $P = 1.1 \times 10^{-8}$). At $P < 1 \times 10^{-5}$, the ONT analysis revealed 62 novel loci, such as *SEPTIN9* (chr17-77351321-77351322, $\beta = -0.27$, $P = 8.4 \times 10^{-6}$) and *TERF2* (chr16-69398923-69398924, $\beta = 0.12$, $P = 8.7 \times 10^{-6}$). Complete results are available in **Supplementary Data 5**. “

Discussion, Page 15, line 4: “The findings from the new generation sequencing EWASs, which were less well-powered, also underscored the role of smoking in carcinogenesis and disrupted neurodevelopment. Previously unreported significant loci (at a threshold of $P < 1 \times 10^{-5}$) identified in the TWIST EWAS included sites mapping to *TSPAN5*, which regulates tumour suppressor gene expression²⁹; *USP42*, involved in head and neck cancer pathogenesis³⁰; and *SST*, encoding somatostatin, a hormone implicated in the development of pancreatic cancer³¹. Significant loci revealed by ONT EWAS included CpGs associated with *SEPTIN9*, a tumour suppressor gene³²; *TERF2*, a telomeric protein linked to tumour formation and progression³³; and *CNTNAP2*, a potential marker of tumour aggressiveness in oligodendrogliomas³⁴, which is also implicated in neurodevelopmental disorders. “

5. How was the batch effect handled in the EWAS? Was it adjusted in the linear regression models as categorical variables to calculate the residuals of pack years? If the cell type proportions were not available, is it possible to estimate (PMID: 22568884)?

We applied linear regression to account for batch effects before conducting the EWAS, removing batch-related variability from the methylation data.

We estimated cell proportions and compared the DNA methylation EWAS with adjustment for estimated blood cell proportions to an unadjusted run. Several high-confidence CpG sites associated with smoking—such as those mapped to *AHRR*, *GPR15*, and *PRSS23*—remained after adjusting for

white blood cell (WBC) proportions. The main differences were minor shifts in PIP values and effect sizes for certain CpG sites, as well as three analysis-specific sites. Specifically, the WBC-adjusted results include two sites with PIP > 0.8 (e.g., cg18146737 mapped to *GFI1* and cg14753356 without a gene annotation) that were absent in the unadjusted analysis (PIPs of 0.98 and 0.90, respectively). Conversely, the unadjusted analysis includes one site (cg08038054 mapped to *GNG11*) that does not appear in the WBC-adjusted results. The results are reported in **Supplementary Data 17**.

Page 25, line 9: “As BayesR implicitly corrects for confounding effects without requiring a full characterization of all covariates, our models were only adjusted for measured variables i.e., estimated white blood cell proportions were not included. However, we conducted a sensitivity analysis by running an EWAS that included adjustments for estimated cell proportions. The results of this sensitivity analysis showed a strong overlap with the primary findings and are presented in **Supplementary Data 17**.”

6. Are X-chromosome data available for the EPIC/TWIST EWAS? If yes, could the authors consider including the analysis results?

Yes, X chromosome data are available for the EPIC, TWIST, and ONT analyses. However, we opted not to include it due to the complexities of X-inactivation and imprinting, which lead to sex-specific methylation patterns and confounding by sex. Additionally, X chromosome data require specialized normalization and statistical adjustments, which could reduce power and introduce bias. For these reasons, we decided to focus on autosomal results, but we acknowledge the value of X-linked analyses for future work.

7. How does the blood-based EWAS result under the Bayesian framework compare with the previous studies that used Illumina 450k BeadChip array (N=15,907, PMID: 27651444) and Illumina EPIC 850k array (N=15,014, PMID: 38199042)? Both these two previous investigations used frequentist-based methods. A search of PIP in the EPIC EWAS results for the previously known CpGs sites would be very helpful as a replication.

It is an interesting question. We have updated **Supplementary Data 2** (listing results of Bayesian EWAS) to include two new columns: “Found in previous Illumina 450k EWAS?” and “Found in previous Illumina EPIC EWAS?”. We focused on sites with significance at $P < 1 \times 10^{-5}$ with exact P values specified in a column called “EWAS catalog P”.

For the study that used Illumina 450k BeadChip array (N=15,907, PMID: 27651444), 19 sites overlapped with our findings at a posterior inclusion probability (PIP) > 80%. The Bayesian approach identified 23 sites not detected by the frequentist method, while the frequentist approach found 9,192 associations that were not observed by the Bayesian method.

Similarly, for the meta-analysis based on Illumina EPIC 850k array (N=15,014, PMID: 38199042), 33 sites showed overlap at a PIP > 80%. The Bayesian approach identified 9 sites not found by the frequentist approach, and the frequentist method detected 21,990 associations not identified by the Bayesian approach.

8. For the EPIC EWAS, the authors performed a comprehensive search in the EWAS Catalog – can the authors expand the discussion of the novel genes identified, such as SCAMP5, FGF20, and SKI? SCAMP5 and FGF20 are involved in neurodevelopment pathways and may affect the brain's response to nicotine and influence addictive behaviors.

We have added the following lines to the Results and Discussion to incorporate these points:

Page 6, line 12: “Forty-two independent CpGs were associated with smoking at posterior inclusion probability (PIP) > 80%, with 26 of these associations reaching a PIP > 95% (**Supplementary Data 2**). Among the associations with PIP > 80%, 33 had previously been reported in the EWAS catalogue¹² at $P < 1 \times 10^{-4}$, with 30 of these reaching $P < 1 \times 10^{-7}$. This catalogue is a resource that curates findings from published EWAS studies.

Novel associations included 9 sites at PIP > 80% and two CpGs with PIP > 95%. The former group included intergenic CpGs linked to neurodevelopment and addiction, such as cg22454588 (annotated to *SCAMP5*), cg27110277 (*FGF20*), and cg19404444 (*SKI*). The latter, high confidence associations, included cg02517189 (*GRIK5*) and cg00562553 (*HOXA4*). “

Page 14, line 14: “Among the novel loci identified in the EPIC array EWAS of smoking at PIP > 80%, five array-based CpGs are annotated to *FGF20*, *SCAMP5*, *GRIK5*, *SKI*, and *HOXA4*. *FGF20* plays a key role in the survival and function of dopamine-producing neurons²², which are crucial in the brain's reward system²³ that nicotine stimulates. *SCAMP5* is essential for dopamine release²⁴ and the pleasurable sensations associated with smoking. It reinforces smoking behaviour and potentially make individuals more susceptible to nicotine addiction. *GRIK5* encodes a glutamate receptor²⁵ which modulates dopaminergic neurons within the brain's reward pathways, influencing how strongly these pathways respond to nicotine²⁶. A proto-oncogene called *SKI* regulates cell growth and apoptosis²⁷. It could potentially modify neural circuitry related to addiction, which affect the risk of developing nicotine dependence or influence the severity of addiction. Lastly, *HOXA4* is a homeobox domain gene that is normally involved in embryonic development²⁸. Homeobox genes are abnormally expressed in cancer cells and changes in the expression of *HOXA4* has been specifically associated with colorectal, ovarian and lung cancer²⁸.”

9. The TWIST EWAS is particularly interesting due to its high coverage of CpG sites. Given the small proportion of overlap compared with EPIC, most CpG sites are novel. A pathway enrichment analysis would be of interest as it may reveal novel mechanisms in addition to existing understanding.

We have conducted a pathway enrichment analysis using TWIST- and ONT- EWAS output at $P < 1 \times 10^{-5}$ and added the results to the manuscripts.

Page 26, line 14: “Gene names associated with CpG sites reaching a suggestive significance threshold ($P < 1 \times 10^{-5}$) were extracted and subjected gene set enrichment analysis in Functional Mapping and Annotation (FUMA) GENE2FUNC tool⁶¹, which implements a hypergeometric test. An FDR-adjusted p-value threshold of 0.05 was applied, and a minimum of 2 overlapping genes within each gene set was required.”

Page 8, line 10: “A gene set enrichment analysis of genes mapped to the 102 CpGs with $P < 1 \times 10^{-5}$ identified in the TWIST EWAS revealed 13 enriched gene sets (FDR $p < 0.05$; see **Supplementary Data 6**). These included tissue degradation through altered extracellular matrix dynamics and chronic inflammation stemming from dysregulated ATP release and immune cell recruitment. Many of the enriched pathways were driven by the presence of collagen genes, such as *COL4A4* and *COL4A3*. In contrast, enrichment analysis based on the significant CpGs from the ONT EWAS did not identify any significantly enriched gene sets.”

Page 15, line 13: “An enrichment analysis of the TWIST results indicated that smoking influences methylation patterns across genes involved in extracellular matrix interactions, chronic inflammation, platelet activation, and ATP regulation, among other pathways. Again, the genes present in enriched pathways typically included *COL4A3* and *COL4A4* which play crucial roles in cancer invasion and metastasis³⁵; and inflammation and remodelling of the lung extracellular matrix³⁶.”

10. For the training of mCigarette, can the authors explain the reason of choosing elastic net over other machine learning methods such as LASSO or ridge regression? A total of 17865 CpG sites were included out of the total 18760 identified in Joehanes et al. (PMID: 27651444), with 895 removed from the algorithm. How many of the CpG sites in EpiSmokEr, BayesR and McCartney et al (GrimAge algorithm not available) overlap with mCigarette? Identifying overlap helps in understanding the consistency and robustness of CpG sites associated with smoking across different studies and methodologies. Non-overlapping sites could suggest additional information or potentially new markers captured by mCigarette, which might contribute to enhancing predictive performance or understanding the biological mechanisms further.

We considered four approaches for training mCigarette:

- a) BayesR EWAS posterior weights,
- b) lasso,
- c) ridge regression,
- d) elastic net.

For b-d, we pre-filtered the input features to consider CpGs significant in the Joehanes et al. meta-analysis. We ultimately chose elastic net, as it produced a predictor that yielded the highest incremental R^2 in linear models of pack years adjusted for age and sex ($\text{pack_years} \sim \text{age} + \text{sex} + \text{mCigarette}$). Please see the table below:

Table 1. *Performance of different machine learning methods used to derive mCigarette. The difference between variance explained by null and full models is denoted as the incremental R^2 . While the null model was adjusted for age and sex, the full model also included the studied score. Pearson’s correlation coefficient is referred to as r .*

Metric / Method	BayesR	Lasso	Ridge	Elastic Net
Incremental R^2	0.406	0.533	0.497	0.534
r	0.646	0.749	0.724	0.750

The following number of CpG sites overlapped between mCigarette and:

- EpiSmokEr (48 sites)
- McCartney et al (19 sites)
- BayesR (279 sites)

955 sites were included only in mCigarette.

This information is now provided in **Supplementary Data 7**.

11. The authors stated that the performance of mCigarette in prediction among adults with mean age 29 and 50 years was excellent, which may not fully capture the findings across age groups. In Supplementary Table 7, which can be converted into a color-coded bar plot for better illustration, we can see that the performance in the antenatal group (age 29) is the best compared with all other age groups. The antenatal group is 100% female with data collected at 18 and 32 weeks gestation, with only 98 current smokers out of 968 participants. This is an important finding as it may suggest that the epigenetic markers used in the model are highly sensitive to smoking-related changes during pregnancy. Can the authors add this to the discussion section and explore the potential implications?

We would like to thank Reviewer #2 for highlighting these. We agree that the strong predictive performance in the antenatal group, which consists entirely of women with data collected during pregnancy, is a noteworthy finding.

We have added **Supplementary Fig. 2** (visualising previous **Supplementary Table 7**) and the following paragraph to the discussion:

Page 15, line 22: “The robust performance of mCigarette among pregnant women in ALSPAC may reflect the unique biological context of pregnancy, where hormonal changes and epigenetic plasticity make smoking-related epigenetic changes more pronounced. Loci which are most responsive to smoking during pregnancy could provide insights into changes transmitted to the foetus, and affecting the child’s health later in life. In the future, mCigarette could be used to monitor smoking cessation efforts during pregnancy, ensuring compliance with cessation programs and potentially improving their effectiveness.”

12. For the DNAm analysis in the 5 brain regions among LBC1936, did the authors consider a sensitivity analysis treating smoking as a continuous variable (pack years)? I wonder if a 1df test would be better powered compared to a 2df test using a categorical variable of smoking status.

Thank you for the suggestion. We have already addressed smoking as a continuous variable by modelling it as an integer in our analysis, rather than as a categorical factor. Specifically, we ran the model as follows: $CpG \sim \text{smoking category}$, where *smoking category* was coded as 0, 1, or 2, representing levels of exposure but treated as a continuous integer rather than a factor. This effectively allowed us to use a 1 df test to capture smoking’s linear association, enhancing our power relative to a categorical approach. We have added the following clarification to the manuscript:

Page 28, line 6: “Using blood-DNAm measured at wave 1 and brain-DNAm data, exploratory EWAS analyses were performed ($CpG \sim \text{smoking category}$). In these analyses, smoking was treated as a continuous variable encoded as 0 = never smoker, 1 = former smoker, 2 = current smoker.”

13. The inflation factor and the LD Score regression intercept for the GWAS results should be reported, as relatedness was addressed in the statistical model in the family-based cohort. The overlap of lead loci compared between the GrimAge DNAm pack years GWAS and smoking pack years GWAS can be visualized in scatterplot of the beta coefficients in addition to the current Figure 5. What's the correlation between smoking pack years and GrimAge-based DNAm estimator of smoking pack years in GS? Can the authors discuss the potential reasons for the relatively moderate genetic correlation of 0.51 between the two phenotypes?

We added the inflation factor and the LD Score regression intercept to the Results section:

Page 12, line 7: "In 17,105 GS individuals, there was a single nucleotide polymorphism (SNP)-based heritability of 27.3% ($P=7.0 \times 10^{-46}$) for self-reported pack years of smoking compared to 41.0% ($P=8.2 \times 10^{-98}$) for GrimAge DNAm pack years (**Figure 5**). The genomic inflation factors (λ) were 1.06 and 1.10 for pack years and GrimAge DNAm pack years, respectively."

Page 13, line 13: "The genetic correlation (r_g) between GrimAge DNAm pack years and pack years from Erzurumluoglu *et al.*²⁰ was 0.62 (SE=0.12, $P=4.4 \times 10^{-7}$), with an LD score regression intercept of 0.99 (SE=0.01). The r_g between self-reported pack years in the 17,105 GS and pack years from Erzurumluoglu *et al.* ($n=131,892$) was 0.67 (SE=0.19, $P=5.0 \times 10^{-4}$), with an LD score regression intercept of 0.99 (SE=0.01)."

Regarding the correlation between smoking pack years and the GrimAge-based DNAm estimator of smoking pack years in GS, we calculated this correlation to be 0.65. We included this information in the revised manuscript:

Page 13, line 12: "GrimAge DNAm pack years and self-reported pack years were moderately correlated (Spearman's $r = 0.65$)."

We agree that a scatterplot of the beta coefficients would be a valuable addition, allowing for a clearer comparison between the two sets of results. We added it as a **Supplementary Fig. 5**.

Page 12, line 22: "The beta coefficients of the lead loci identified in the GrimAge DNAm pack years GWAS and the smoking pack years GWAS are compared in **Supplementary Fig. 5**."

As for the moderate genetic correlation between epigenetic and self-reported smoking, we have added the following explanation to the Discussion section:

Page 16, line 14: "In GS, the GrimAge DNAm pack years GWAS results did not align with self-reported smoking GWAS findings but did show partial overlap of lead loci with the most extensive GWAS of self-reported smoking to date. This may suggest an increased power to detect significant loci when the epigenetic score is analysed as a phenotype. However, the genetic correlation between GrimAge DNAm pack years and the meta-analysis smoking pack years was considerably lower than the latter and self-reported pack years in GS. The moderate correlation could reflect differences in the biological pathways that phenotypic and epigenetic measures of smoking are capturing. The GrimAge-based DNAm estimator is designed to capture the cumulative biological impact of smoking, which might include broader aging-related processes beyond direct tobacco exposure. In contrast, the smoking pack years represent a more straightforward measure of cumulative smoking exposure."

14. In Supplemental Tables 9-13, the Genome Reference Consortium Human Build version should be specified. Seems like the usage of GRCh37 and GRCh38 is not consistent. Could this help address the missingness of SNPs while comparing across GWAS summaries (Erzurumluoglu et al. was GRCh37, Saunders et al., 2022 was GRCh38)? By briefly searching the SNPs from the Erzurumluoglu et al. summary (https://ftp.ebi.ac.uk/pub/databases/gwas/summary_statistics/GCST007001-GCST008000/GCST007601/), some SNPs labeled as missing are available.

We have brought all summary statistics to the same Genome Reference Consortium Human Build version (GRC37), with notation added to the Supplemental Tables, and re-run the relevant genetic correlation analyses (please see pt 12.). The differences are very slight and likely have no practical impact on the interpretation of genetic correlations between traits in the study.

The missing SNPs from previous **Supplemental Table 9** (now **Supplementary Data 10**) were initially filtered out from the Erzurumluoglu et al. summary statistics during the early stages of data pre-processing (as all had $P > 0.05$). We have now reintroduced them.

Minor issues:

Page 10, Line 15, first appearance of “FOM/FOF” should be specified.

We specified the first appearance of “FOM/FOF”.

Page 21, line 14: “Antenatal collection includes data from the ALSPAC mothers only, the Focus on Mothers (FOM)/ Focus on Fathers (FOF) collection corresponds to the mothers/fathers at midlife (~50 years) ⁴⁹”

Supplemental Table 11, the column names should be explained in the footnote. The “Ptext” column can be deleted if all values are not available.

We have explained the column names in the footnote of **Supplemental Table 11** (now **Supplementary Data 12**). We have removed the Ptext column.

Supplemental Table 12, the column name “CisTrans” and its coding should be added to the footnote.

We have added footnote to **Supplementary Data 13** explaining the meaning of the CisTrans column and detailing its coding: “CisTrans - 1 indicates that the association is *cis* (<1 Mb from the DNAm site), 0 indicates that it is *trans*.”

Reviewer #3 (Remarks to the Author):

Chybowska et al. used several approaches to characterize smoking-associated DNA methylation changes in humans. In the Generation Scotland Cohort, Illumina EPIC850K arrays (n=17,865) and TWIST methylome panels (n=48) were used to conduct epigenome-wide association studies (EWASs) on DNA methylation in whole blood. The Generation Scotland data were also used to derive a composite biomarker of cigarette smoke exposure (mCigarette), which was then evaluated against other composite biomarkers of cigarette exposure using DNA methylation data from two independent cohorts (Lothian Birth Cohort of 1936 and ALSPAC). DNA methylation data from whole blood and 5 brain regions of 14 individuals from the Lothian Birth Cohort of 1936 were used to examine tissue-specific effects. In my opinion, the size of the cohort used in the EWAS analysis and

in the development of a biomarker of smoking. along with the inclusion of two independent cohorts to evaluate use of mCigarette, contribute to further the understanding of the association between cigarette smoking and DNA methylation in humans.

We would like to thank Reviewer #3 for their comprehensive feedback on our work. The recognition of the cohort size, inclusion of two independent cohorts and the fact that we conducted a multi-tissue analysis as strengths of our study are encouraging. We believe these aspects significantly enhance the understanding of the complex relationship between cigarette smoking and DNA methylation. We are grateful for the insights, which have further motivated us to continue our research in this area. We have added our responses to comments below, in blue.

Major Comments:

1. The methods are somewhat unclear as to which DNA methylation datasets represent newly generated data versus reanalysis of previously published datasets. Although this information is provided in the Reporting Summary, authors should update methods to make this distinction more obvious in the text.

All LBC1936 and ALSPAC analyses were secondary in nature, in addition to the GS EPIC analyses.

The generated data included TWIST Bioscience and Oxford Nanopore Technology sequencing (ONT) DNAm for 24 pairs of smokers and non-smokers from Generation Scotland. The ONT data were added to the manuscript as part of this revision. We have made the following changes in the Methods section.

Page 23, line 19: “Whole methylome sequencing data were generated for 48 unrelated smokers and non-smokers from GS as part of this study. Two approaches were used: the TWIST methylome panel (~4 million CpG sites) and ONT sequencing (~21 million CpG sites).”

Additional changes related to ONT data:

Abstract: “In this work, we conducted a Bayesian Epigenome-Wide Association Study of smoking pack years (n=17,865, ~850k sites, Illumina EPIC array) and extended it by analysing whole genome data of smokers and non-smokers from Generation Scotland (n=46, ~4–21 million sites via TWIST and Oxford Nanopore sequencing).”

Page 24, line 4: “Sequencing using the TWIST Human Methylome Panel was performed by the Genetics Core, Edinburgh Clinical Research Facility according to TWIST Targeted Methylation Sequencing Protocol⁵³. Sequencing using the ONT kit was performed by Edinburgh Genomics (the first 24 libraries, without basecalling) and the Genetics Core, Edinburgh Clinical Research Facility on the Oxford Nanopore PromethION 24, with R10.4.1 flow cells, running for 72 hours. Further details available in **Supplementary Methods**.

Dorado, optimized for NVIDIA GPUs, was used for high-accuracy basecalling and modified base detection in raw ONT data. Reads were aligned to GRCh38 human reference genome. Variants were called with epi2me-labs/wf-human-variation nextflow pipeline, version 23.10.1. Methylation level and depth of coverage was measured at 28,989,402 covered CpG sites.

The bedGraph (TWIST) and bedMethyl (ONT) files were subsequently processed in R version 4.3.1⁵⁶ using Methrix package⁵⁷. As part of post-processing, loci a) of extremely low (minimal DoC = 2) and

high coverage (beyond 0.99 quantile), and b) overlapping with known cytosine to thymine polymorphisms were removed from the methylation dataset. Finally, a coverage filter was applied, retaining only the loci that were covered in at least 40 samples by: a) 10 or more reads (TWIST), b) 5 or more reads (ONT). This left an analysis sample of 3,391,718 and 21,167,712 CpGs for TWIST and ONT data, respectively. CpG sites were annotated with Annotatr R package⁵⁸. “

Page 26, line 6: “DNAm of 24 pairs of smokers and non-smokers from GS was profiled using the EPIC array, ONT kit, and TWIST platform (see Methods - Sequencing-based approach). During quality control, one pair was identified as mismatched due an age gap exceeding the pre-defined threshold of 12 months. To preserve the integrity of the study design and prevent potential biases in the analysis, this pair was excluded from subsequent analyses. This left an analysis sample of 46 individuals.”

Page 7, line 3: “Next, we extended this analysis by running high resolution EWASs of smoking on a subset of 23 pairs (n=46) of current vs never smokers with Illumina EPIC array (~850k CpG sites), TWIST human methylation panel (~4 million CpG sites), and Oxford Nanopore Technologies (ONT) sequencing data (~21 million CpG sites). At $P < 3.6 \times 10^{-8}$, the EPIC-based analysis revealed 15 CpG sites associated with smoking status (EWAS inflation factor, $\lambda = 0.94$), while the TWIST-based and ONT-based analyses identified 33 ($\lambda = 1.60$) and 9 ($\lambda = 1.11$) associations, respectively. At a less stringent threshold ($P < 1 \times 10^{-5}$), these counts increased to 42, 102, and 63 for the EPIC-, TWIST-, and ONT-based analyses, respectively. The overlap between the sites identified by these technologies is detailed in **Supplementary Data 3. Figure 2** and **Supplementary Fig. 1** (comparison of beta estimates) display the results obtained from the TWIST, ONT and EPIC EWAS.

Among the 33 associations identified as significant in the TWIST EWAS at $P < 3.6 \times 10^{-8}$, two had been previously reported in the EWAS catalog (based on DNAm profiled with array technologies). These included *AHRR* (chr5-373263-373264, $\beta = -0.35$, $P = 1.2 \times 10^{-10}$) and an intergenic locus found on chromosome 2 (chr2-232419951-232419952, $\beta = -0.24$, $P = 3.5 \times 10^{-8}$). The remaining 31 loci significant at this threshold were novel, including sites annotated to *F2RL3* (chr19-16889741-16889742, $\beta = -0.31$, $P = 1.3 \times 10^{-11}$) and *USP42* (chr7-6126706-6126707, $\beta = 0.05$, $P = 1.8 \times 10^{-8}$). At a less stringent threshold of $P < 1 \times 10^{-5}$, 98 sites were identified as novel, including *SST* (chr3-187670342-187670343, $\beta = -0.09$, $P = 2.2 \times 10^{-7}$) and *TSPAN5* (chr4-98472405-98472406, $\beta = -0.06$, $P = 6.8 \times 10^{-6}$). Further details are provided in **Supplementary Data 4.**”

Results, Page 8, line 1: “In the ONT EWAS, 9 sites were significant at $P < 3.6 \times 10^{-8}$, of which only one had been previously catalogued: a site mapping to *AHRR* (chr5-373263-373264, $\beta = -0.47$, $P = 2.4 \times 10^{-12}$). The remaining eight were novel, including additional loci within the *AHRR* region, loci from an intergenic region on chromosome 2 (e.g., chr2-232420079-232420080, $\beta = -0.37$, $P = 3.4 \times 10^{-10}$) and a site annotated to *CNTNAP2* (chr7-147245588-147245589, $\beta = 0.37$, $P = 1.1 \times 10^{-8}$). At $P < 1 \times 10^{-5}$, the ONT analysis revealed 62 novel loci, such as *SEPTIN9* (chr17-77351321-77351322, $\beta = -0.27$, $P = 8.4 \times 10^{-6}$) and *TERF2* (chr16-69398923-69398924, $\beta = 0.12$, $P = 8.7 \times 10^{-6}$). Complete results are available in **Supplementary Data 5.** “

Discussion, Page 15, line 4: “The findings from the new generation sequencing EWASs, which were less well-powered, also underscored the role of smoking in carcinogenesis and disrupted neurodevelopment. Previously unreported significant loci (at a threshold of $P < 1 \times 10^{-5}$) identified in the TWIST EWAS included sites mapping to *TSPAN5*, which regulates tumour suppressor gene expression²⁹; *USP42*, involved in head and neck cancer pathogenesis³⁰; and *SST*, encoding somatostatin, a hormone implicated in the development of pancreatic cancer³¹. Significant loci revealed by ONT EWAS included CpGs associated with *SEPTIN9*, a tumour suppressor gene³²; *TERF2*, a telomeric protein linked to tumour formation and progression³³; and *CNTNAP2*, a potential marker of tumour aggressiveness in oligodendrogliomas³⁴, which is also implicated in neurodevelopmental disorders. “

2. Cell composition of whole blood has been demonstrated to be affected by a variety of factors including age, smoking status, and immunological history (Bergstedt et al. 2022). In section 2.5 of Methods (pg. 13, lines 4-6) the authors indicate that EWAS models were not adjusted for estimated white blood cell proportions as “BayesR implicitly corrects for confounding effects.” Given the 1) substantial impact that variables not included in this study (e.g., cytomegalovirus serostatus) have on DNA methylation and composition of immune cells in blood and 2) the cell-type specific, smoking-associated DNA methylation changes reported in (Wang et al. 2023), I recommend comparing the DNA methylation EWAS with adjustment for estimated blood cell proportions to results presented in section 3.1 of Results.

We appreciate this point; Reviewer #2 was also interested in this analysis. We estimated cell proportions and compared the DNA methylation EWAS with adjustment for estimated blood cell proportions to an unadjusted run. Several high-confidence CpG sites associated with smoking—such as those mapped to *AHRR*, *GPR15*, and *PRSS23*—remained after adjusting for white blood cell (WBC) proportions. The main differences were minor shifts in PIP values and effect sizes for certain CpG sites, as well as three analysis-specific sites. Specifically, the WBC-adjusted results include two sites with PIP > 0.8 (e.g., cg18146737 mapped to *GFI1* and cg14753356 without a gene annotation) that were absent in the unadjusted analysis (PIPs of 0.98 and 0.90, respectively). Conversely, the unadjusted analysis includes one site (cg08038054 mapped to *GNG11*) that does not appear in the WBC-adjusted results. The results are reported in **Supplementary Data 17**.

Page 25, line 9: “As BayesR implicitly corrects for confounding effects without requiring a full characterization of all covariates, our models were only adjusted for measured variables i.e., estimated white blood cell proportions were not included. However, we conducted a sensitivity analysis by running an EWAS that included adjustments for estimated cell proportions. The results of this sensitivity analysis showed a strong overlap with the primary findings and are presented in **Supplementary Data 17**.”

3. In the abstract, the authors acknowledge that methylation levels at several CpG sites show “near-perfect discrimination of smoking status” in blood (pg. 3, line 7). Whereas they describe the ability of the mCigarette biomarker to discriminate across smoking categories as “excellent” (pg. 3, line 4). I think it is crucial to benchmark any composite DNA methylation biomarkers of smoking against CpG sites that have been reproducibly demonstrated to discriminate amongst categories of smoking status in multiple cohort studies, specifically DNA methylation at cg05575921 (*AHRR*) in blood samples. As such, it would be beneficial to include a comparison using cg05575921 as a biomarker of smoking in the evaluation of mCigarette (Results 3.3., Table 1).

We agree that benchmarking the mCigarette biomarker against well-established individual CpG sites, such as cg05575921 at *AHRR*, would provide valuable context for evaluating its performance. cg05575921 has indeed been widely recognized for its strong association with smoking status across multiple studies. We included a comparison using cg05575921 as a single-site biomarker of smoking in the evaluation of mCigarette, as suggested (**Table 1**). We found that the *AHRR* single-site biomarker provided a robust and highly practical option for smoking classification, particularly in settings where limited data points or resources are available. However, multi-site models offered greater precision, accounting for a broader smoking-related methylation signature.

Table 1. description: “A single-site biomarker for smoking, based on methylation at *AHRR* (cg05575921), was also trained in the same subset of GS individuals (n = 17,865) using linear regression. Both mCigarette and the single-site biomarker were tested in wave 1 of LBC1936 (n = 882, mean age 70 years), while mCigarette alone was tested in ALSPAC (n = 496–1,207 across four time points). These biomarkers were benchmarked against three epigenetic scores for smoking: EpiSmokEr score^{7,35,36}, and two scores derived from previous GS analyses on smaller subsets of the dataset - one based on BayesR weights³¹, the other via lasso penalised regression by McCartney *et al.*¹⁰. In LBC1931, the predictive performance of mCigarette was additionally compared to that of GrimAge DNAm pack years³⁷.”

Table 1:

Metric	AHRR	EpiSmokEr	BayesR	McCartney et al	GrimAge	mCigarette
N training	17,865	1,793	9,448	5,087	1,731	17,865
a) Variance explained in measured pack years (R^2) and correlation (r) metrics						
Incremental R^2	0.329	0.351	0.514	0.330	0.419	0.534
r	0.589	0.610	0.735	0.581	0.664	0.750
b) Binary classification performance (AUC)						
Current / Never	0.972	0.985	0.977	0.982	0.983	0.984
Current / Former	0.930	0.916	0.853	0.921	0.939	0.897
Former / Never	0.742	0.755	0.846	0.725	0.815	0.852

Page 10, line 1: “The predictive performance of mCigarette was benchmarked against four previously developed epigenetic scores for smoking, as well as a novel single-site biomarker based on *AHRR* methylation at cg05575921, in wave 1 of LBC1936 (n=882).”

Page 16, line 4: “The *AHRR* single-site biomarker provided a highly practical option for smoking classification, particularly in settings where limited data points or resources are available. However, multi-site models offered greater precision, accounting for a broader smoking-related methylation signature.”

Minor Comments:

1. Pg. 3, line 7: Change “both” to “either”

This sentence was removed from the manuscript to meet the Editorial requirement of a maximum 150-word abstract.

2. Pg. 5, line 3: Change “molecule” to “molecules”

We changed “molecule” to “molecules”.

Page 4, line 2: “DNA methylation (DNAm) is a cell- and tissue-specific epigenetic modification of DNA molecules that does not change the DNA sequence itself”

3. Pg. 9, lines 20-22: This sentence indicates the whole blood and brain tissue samples were measured with “HumanMethylation450 BeadChip,” whereas, Figure 1 (pg. 7) indicates “EPIC EWAS” and reference to the DNA methylation data cited in line 22 (Stevenson et al. 2022) indicates that Illumina 850K arrays were used to quantify DNA methylation in brain samples and Illumina 450K arrays were used to profile the blood samples for the Lothian Birth Cohort.

We are grateful to you for spotting this issue. We have added a clarification.

Page 20, line 13: “DNAm from whole blood has been measured using Illumina Infinium HumanMethylation450 BeadChip array, while DNAm in five post-mortem brain tissues was profiled using the Illumina EPIC850k array⁴⁴”

4. Pg. 14, line 7-8: This sentence indicates that results from TWIST and EPIC850k arrays were “compared on a Manhattan plot.” Please change to “displayed on Manhattan plots” as the separate Manhattan plots in Figure 2. (page 20) do not compare the results of the two methods; they merely display the results from each analysis near each other.

We made this change.

Page 26, line 14: “The results were displayed on a Manhattan plot generated with the CMplot R package⁶⁰”

5. Pg. 19, line 5-7: Were all 44 associations from the EPIC-based analysis evaluated in the TWIST-based analysis? That is, were they just not significant at $P < 1 \times 10^{-5}$ in the TWIST analysis or were they excluded for technical reasons (e.g., not passing QC)? Clarification on this point in the manuscript text is recommended.

We have added an additional supplemental table (Supplemental Table 6) addressing this question:

Page 7, line 7: “At $P < 3.6 \times 10^{-8}$, the EPIC-based analysis revealed 15 CpG sites associated with smoking status (EWAS inflation factor, $\lambda = 0.94$), while the TWIST-based and ONT-based analyses identified 33 ($\lambda = 1.60$) and 9 ($\lambda = 1.11$) associations, respectively. At a less stringent threshold ($P < 1 \times 10^{-5}$), these counts increased to 42, 102, and 63 for the EPIC-, TWIST-, and ONT-based analyses, respectively. The overlap between the sites identified by these technologies is detailed in **Supplementary Data 3.**”

6. Pg. 19, line 9: Change “presents a comparison of” to “displays.”

Thank you, we made this change!

Page 7, line 11: “Figure 2 and Supplementary Fig. 1 (comparison of beta estimates) display the results obtained from the TWIST, ONT and EPIC EWAS.”

7. Pg. 19, line 12-13: In this sentence, I think it would be informative to include the number of associations that had previously been reported and the number of loci that were novel (as opposed to stating that some had been reported and some were novel).

Thank you for the suggestion. Previously, we conducted an EWAS catalog lookup by gene name, as cg identifiers were unavailable for TWIST data and the chromosomal coordinates in the EWAS catalog were incompatible with our reference genome version (hg19). Now, we have lifted the hg19-based coordinates from the EWAS catalog to hg38 and conducted the search by chromosomal position.

Page 7, line 15: “Among the 33 associations identified as significant in the TWIST EWAS at $P < 3.6 \times 10^{-8}$, two had been previously reported in the EWAS catalog (based on DNAm profiled with array technologies). These included *AHRR* (chr5-373263-373264, $\beta = -0.35$, $P = 1.2 \times 10^{-10}$) and an intergenic locus found on chromosome 2 (chr2-232419951-232419952, $\beta = -0.24$, $P = 3.5 \times 10^{-8}$). The remaining 31 loci significant at this threshold were novel, including sites annotated to *F2RL3* (chr19-16889741-16889742, $\beta = -0.31$, $P = 1.3 \times 10^{-11}$) and *USP42* (chr7-6126706-6126707, $\beta = 0.05$, $P = 1.8 \times 10^{-8}$). At a less stringent threshold of $P < 1 \times 10^{-5}$, 98 sites were identified as novel, including *SST* (chr3-187670342-187670343, $\beta = -0.09$, $P = 2.2 \times 10^{-7}$) and *TSPAN5* (chr4-98472405-98472406, $\beta = -0.06$, $P = 6.8 \times 10^{-6}$). Further details are provided in **Supplementary Data 4.**”

We have also conducted a similar analysis for ONT data:

Page 8, line 1: “In the ONT EWAS, 9 sites were significant at $P < 3.6 \times 10^{-8}$, of which only one had been previously catalogued: a site mapping to *AHRR* (chr5-373263-373264, $\beta = -0.47$, $P = 2.4 \times 10^{-12}$). The remaining eight were novel, including additional loci within the *AHRR* region, loci from an intergenic region on chromosome 2 (e.g., chr2-232420079-232420080, $\beta = -0.37$, $P = 3.4 \times 10^{-10}$) and a site annotated to *CNTNAP2* (chr7-147245588-147245589, $\beta = 0.37$, $P = 1.1 \times 10^{-8}$). At $P < 1 \times 10^{-5}$, the ONT analysis revealed 62 novel loci, such as *SEPTIN9* (chr17-77351321-77351322, $\beta = -0.27$, $P = 8.4 \times 10^{-6}$) and *TERF2* (chr16-69398923-69398924, $\beta = 0.12$, $P = 8.7 \times 10^{-6}$). Complete results are available in **Supplementary Data 5.**”

8. Pg. 23, line 1: Delete “in”

Page 9, line 1: “The predictive performance of mCigarette ~~is~~ was benchmarked against four previously developed epigenetic scores for smoking, as well as a novel single-site biomarker based on *AHRR* methylation at cg05575921, in wave 1 of LBC1936 (n=882).”

9. Pg. 25, line 12: Should be cg05575921

Thank you, we corrected it!

Page 11, line 11: “For instance, the methylation status at cg05575921, annotated to the *AHRR* gene, is a well-established marker of smoking status in whole-blood DNAm.”

10. Pg. 25, line 16: Change “brain tissue samples” to “hippocampus samples” unless this finding (cg26381592) can be generalized to other brain regions tested.

We made this change.

Page 11, line 16: “On the other hand, cg26381592, annotated to the *PMS1* gene, did not effectively distinguish smokers in blood samples, but it exhibited a strong correlation with smoking status in hippocampus samples.”

11. Pg. 26, Figure 4: It would be informative to include figures showing DNA methylation at cg05575921 and cg26381592 in the last blood of the individuals that are shown in the hippocampus data.

Thank you, we have included this information as **Supplementary Fig. 3**.

12. Pg. 31, lines 20-21: Include reference for association between genes and neurological conditions and/or lung cancer.

We included this reference.

Page 17, line 3: “Variations in these genes can affect nicotine dependence and may be associated with neurological conditions as well as lung cancer^{38,39}.”

References:

Bergstedt J, Azzou SAK, Tsuo K, Jaquaniello A, Urrutia A, Rotival M, Lin DTS, MacIsaac JL, Kobor MS, Albert ML, Duffy D, Patin E, Quintana-Murci L; Milieu Intérieur Consortium. The immune factors driving DNA methylation variation in human blood. *Nat Commun*. 2022 Oct 6;13(1):5895. doi: 10.1038/s41467-022-33511-6. PMID: 36202838; PMCID: PMC9537159.

Wang X, Campbell MR, Cho HY, Pittman GS, Martos SN, Bell DA. Epigenomic profiling of isolated blood cell types reveals highly specific B cell smoking signatures and links to disease risk. *Clin Epigenetics*. 2023 May 25;15(1):90. doi: 10.1186/s13148-023-01507-8. PMID: 37231515; PMCID: PMC10211291.

Reviewer #3 (Remarks on code availability):

Code was well-organized and easy to follow. I did not install and run the code to evaluate reproducibility. README file provided thorough instructions and appropriate references to code developed by other groups and used in these analyses.